# Leveraging Low-Rank and Sparse Recurrent Connectivity for Robust Closed-Loop Control

**Neehal Tumma**[1]  **Mathias Lechner**[2]  **Noel Loo**[2]  **Ramin Hasani**[2]  **Daniela Rus**[2]
[1]Harvard University   [2]MIT CSAIL

## Abstract

Developing autonomous agents that can interact with changing environments is an open challenge in machine learning. Robustness is particularly important in these settings as agents are often fit offline on expert demonstrations but deployed online where they must generalize to the closed feedback loop within the environment. In this work, we explore the application of recurrent neural networks to tasks of this nature and understand how a parameterization of their recurrent connectivity influences robustness in closed-loop settings. Specifically, we represent the recurrent connectivity as a function of rank and sparsity and show both theoretically and empirically that modulating these two variables has desirable effects on network dynamics. The proposed low-rank, sparse connectivity induces an interpretable prior on the network that proves to be most amenable for a class of models known as closed-form continuous-time neural networks (CfCs). We find that CfCs with fewer parameters can outperform their full-rank, fully-connected counterparts in the online setting under distribution shift. This yields memory-efficient and robust agents while opening a new perspective on how we can modulate network dynamics through connectivity.

## 1 Introduction

Building models that are robust under natural distribution shift has long been a goal in artificial intelligence (Taori et al., 2020). Existing techniques have sought to address this challenge through various approaches, including domain adaptation (Farahani et al., 2020), transfer learning (Zhuang et al., 2019) and data augmentation (Perez & Wang, 2017). However, these techniques come with drawbacks: domain adaptation and transfer learning can be computationally expensive, and data augmentation often suffers from robustness gains that are not uniform across corruption types (Ford et al., 2019). More generally, machine learning systems tend to perform poorly under distribution shift because approaches like these only serve to ameliorate a problem that is rooted in the model architecture itself.

Natural learning systems serve to address this problem by interacting with their environment to understand the world. They make use of biologically-inspired frameworks that account for distribution shifts by modeling temporal structure (Hasani et al., 2020b). These models are particularly well-suited in a paradigm where they are fit on offline, passive datasets using imitation learning, but deployed in closed-loop, active testing settings (de Haan et al., 2019). This domain is commonly referred to as the open-loop to closed-loop causality gap (Lechner et al., 2022) in which generalization requires the agent to learn a coherent representation of its world (Lechner et al., 2020). One particularly effective framework known as closed-form continuous-time neural networks (CfCs) were shown to outperform many state-of-the-art recurrent models in closed-loop settings (Hasani et al., 2021). These models leverage a sparse wiring structure induced at initialization, yielding compact and interpretable networks (Chahine et al., 2023). However, the reason as to why sparse connectivity is useful in closed-loop systems is poorly understood. In this work, we will address this question by proposing and analyzing an interpretable parameterization of connectivity in CfCs.

To do so, we turn to another class of models: low-rank recurrent neural networks (low-rank RNNs). A low-rank RNN is a network whose *recurrent* connectivity matrix is low-rank. Deriving inspiration

from neural activity in the brain, these networks have provably low-dimensional patterns of activity which makes them successful in many simple neuroscience-based tasks (Mastrogiuseppe & Ostojic, 2018). In this paper, we leverage ideas from previous work on CfCs and low-rank RNNs in order to devise a novel parameterization of recurrent connectivity that improves robustness in closed-loop settings and offers interpretable measures for the network dynamics it incites. In doing so, we show the following:

- Parameterizing recurrent connectivity as a function of rank and sparsity yields provable and interpretable network dynamics (Section 3.1)
- Low-rank and sparse recurrent neural networks can outperform their full-rank, fully-connected counterparts under distribution shift in closed-loop environments (Section 4.1)
- Pruning by inducing sparsity and pruning by enforcing a low-rank structure are distinct in the types of network dynamics each induces (Section 4.3)
- CfCs are more amenable to a low-rank and sparse connectivity prior than other canonical recurrent architectures such as LSTMs (Section 4.4)

## 2 RELATED WORK

In this section, we describe previous works that are closely related to the core findings of the paper.

**Robustness in closed-loop settings.** A class of continuous-time recurrent models known as liquid neural networks (Hasani et al., 2020a) have been shown to achieve state-of-the-art performance under distribution shift by explicitly accounting for external interventions to environment conditions. Liquid neural networks are a prime example of natural learning systems that learn robust, and provably causal, representations (Vorbach et al., 2021). CfCs are a specific instance of liquid neural networks that can leverage sparse connectivity to learn robust representations in the closed-loop causality gap setting.

Outside of modifying the model itself, other approaches to training robust models under an imitation learning framework include augmentation strategies, human interventions (Ross et al., 2010), goal-conditioning (Codevilla et al., 2017), reward conditioning (Srivastava et al., 2019), task-embedding (James et al., 2018) and meta-learning (Finn et al., 2017). These advances fail to consider the structure of the underlying policy; in contrast, liquid neural networks improve the robustness of the decision-making process in autonomous agents, leading to better generalization under the same training distribution (Vorbach et al., 2021).

**Relating connectivity and network dynamics.** A long-standing goal in computational neuroscience is to understand the relationship between structure and function in the brain (Sporns, 2013). We can ask the analogous question in the context of artificial neural networks: what role does model connectivity at initialization play in learned network dynamics? Amongst recurrent neural networks, echo state networks (Goodfellow et al., 2016) present one example of a model that induces static sparsity in order to modulate the complexity of dynamics along the recurrent dimension. However, unlike in our case, echo state networks fix their recurrent connectivity during training. Other works have found that sparse networks can yield provably more robust models (Guo et al., 2018) that generalize across many tasks (Chen et al., 2022) outside the

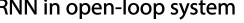

RNN in open-loop system

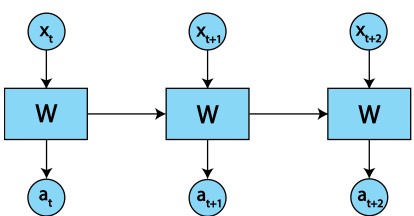

RNN in closed-loop system

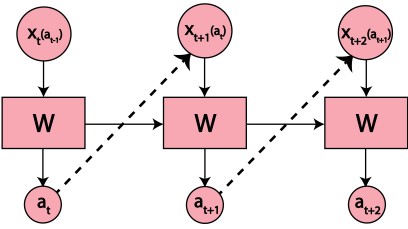

Figure 1: Open-loop systems receive ground-truth observations $x_i$. Closed-loop systems receive observations $x_i(a_{i-1})$ that are a function of the previous actions the agent takes. There is no external feedback to correct the agent in the closed-loop setting.

recurrent setting. With respect to modulating network rank, the advent of the low-rank RNN framework (Mastrogiuseppe & Ostojic, 2018) is one of the first attempts at explicitly modeling the low-dimensional dynamics of the brain. Some follow-ups on this work include examining how low-rank

connectivity arises in full-rank settings (Schuessler et al., 2020) and an analysis of how some tasks are more amenable to low-rank connectivity than others (Dubreuil, 2022). Our work borrows approaches and intuition from many of these prior studies, but is distinct with respect to the proposed connectivity, analysis of network dynamics and task setting.

**Pruning at initialization.** Neural network pruning typically either removes connections in a costly iterative training-retraining paradigm (Frankle & Carbin, 2018) or modifies the objective function to promote sparsity (Goodfellow et al., 2016) which can lead to difficulties in optimization. Recently, pruning at initialization has emerged as an approach to resolve these issues. Pruning at initialization attempts to take a randomly initialized network and remove weights before training (Wang et al., 2021). Various approaches include using connectivity sensitivity (Lee et al., 2018), maintaining dynamical isometry (Lee et al., 2019) and training on supermasks (Ramanujan et al., 2019). Generally, these techniques outperform sparsity induced randomly at initialization. Another approach to pruning is using rank decomposition. One example is low-rank matrix factorization, which is good at reducing size but at the cost of performance (Sainath et al., 2013). The most effective low-rank compression techniques usually leverage a low-rank approximation on the full-rank matrix either during or after training (Xu et al., 2020). In contrast, in this work, we propose a parameterization of connectivity that randomly induces sparsity and leverages a low-rank decomposition of the recurrent weights at initialization – the most straightforward and least costly form of pruning.

**Alternative RNN parameterizations.** Existing works proposed other recurrent architectures such as Lipschitz RNNs (Erichson et al., 2021), antisymmetric RNNs (Chang et al., 2019) and Cayley RNNs (Helfrich et al., 2018) which, like us, propose alternative parameterizations of recurrent connectivity. However, unlike our study, these works introduce changes to the underlying network architecture, whereas our work focuses on the applications of low-rank and sparse recurrent connectivity in more canonical recurrent networks. This is in line with previous works in the closed-loop causality gap setting (Chahine et al., 2023) which also restrict the scope of their models as such.

## 3 PARAMETERIZATION OF CONNECTIVITY

Consider a standard RNN whose input is denoted by $\boldsymbol{x}_t \in \mathbb{R}^n$ and hidden state is given by $\boldsymbol{h}_t \in \mathbb{R}^h$ for time step $t$, hidden size $h$ and input size $n$. Then the functional form of the RNN is given by

$$\boldsymbol{h}_t = \tanh(\boldsymbol{W}_{rec}\boldsymbol{h}_{t-1} + \boldsymbol{W}_{inp}\boldsymbol{x}_t + \boldsymbol{b}) \tag{1}$$

such that $\boldsymbol{W}_{rec} \in \mathbb{R}^{h \times h}$ denotes the recurrent weights, $\boldsymbol{W}_{inp} \in \mathbb{R}^{n \times h}$ denotes the input weights and $\boldsymbol{b}$ denotes the bias. We now propose an alternative parameterization of connectivity for the recurrent weights. Consider a singular value decomposition of $\boldsymbol{W}_{rec}$ given by $\boldsymbol{W}_{rec} = \boldsymbol{U}\boldsymbol{\Sigma}\boldsymbol{V}^T$. The canonical rank $r$ approximation of $\boldsymbol{W}_{rec}$ is $\boldsymbol{U}_r\boldsymbol{\Sigma}_r\boldsymbol{V}_r^T$ such that $\boldsymbol{\Sigma}_r$ consists of the top-$r$ singular values and $\boldsymbol{U}_r$ and $\boldsymbol{V}_r^T$ consist of the corresponding singular vectors. Using this low-rank approximation, we can construct the following parameterization of the recurrent weights for a given rank $r$ and sparsity level $s$:

$$\boldsymbol{W}_{rec}(r,s) = \boldsymbol{W}_1(r)\boldsymbol{W}_2(r) \odot \boldsymbol{M}(s) \tag{2}$$

where $\boldsymbol{W}_1 = \boldsymbol{U}_r(\boldsymbol{\Sigma}_r)^{1/2}$, $\boldsymbol{W}_2 = (\boldsymbol{\Sigma}_r)^{1/2}\boldsymbol{V}_r^T$ and $\boldsymbol{M} \in \mathbb{F}_2$ is a random binary mask such that each entry $\boldsymbol{M}_{ij} \sim 1 - \text{Bernoulli}(s)$. This low-rank and sparse parameterization of $\boldsymbol{W}_{rec}$ is a generalization of the parameterization studied in Herbert & Ostojic (2022) to recurrent connectivity of arbitrary rank and trainable weights $\boldsymbol{W}_1, \boldsymbol{W}_2$ as opposed to restricting ourselves to the setting of rank-1 matrices and fixed weights in which dynamical analysis is more tractable, but the network is constrained in expressivity. Note that the random mask $\boldsymbol{M}$ is fixed throughout training.

### 3.1 NETWORK DYNAMICS AT INITIALIZATION

We propose this parameterization of recurrent connectivity as it allows us to modulate aspects of the network that instill an inductive bias, which proves to be beneficial for performance in the closed-loop setting, particularly under distribution shift. We care about three measures of $\boldsymbol{W}_{rec}(r,s)$ in particular: the spectral radius, the spectral norm and the rate of decay of the singular value spectrum.

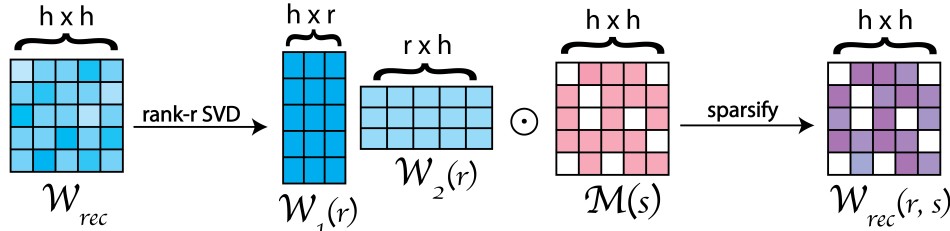

Figure 2: Parameterization of recurrent connectivity as a function of rank and sparsity.

The spectral radius $\lambda_{max}$ of $\boldsymbol{W}_{rec}(r, s)$ is widely accepted as a proxy for the rate at which the gradient evolves backwards in time (Pascanu et al., 2012). Namely, if $\lambda_{max} < 1$, an RNN has a vanishing gradient (refer to A.5.1 for details). In many time series applications, having a constant error flow is a desirable property, as arbitrary data sequences may have long-term relations. However, in the case of many closed-loop systems, learning long-term dependencies can be detrimental in the online setting due to the short-term causality inherent to the task (Lechner et al., 2020). To put this into context, in this work we consider agents learning to play games in various Arcade Learning Environments (ALEs). Agents that perform well in ALEs often use frame stacking (Horgan et al., 2018a) which enforces a strict short-horizon temporal prior by leveraging a look-back window of only a few input frames. Hence, networks like LSTMs or GRUs may capture spurious long-term dependencies and thus learn inadequate models. In contrast, the vanishing of gradients that tends to be more pronounced in RNNs counterintuitively enhances the performance of the agent, as it places a short-horizon prior on the temporal attention span of the network.

We will motivate the importance of the singular values in the distribution shift setting. Consider an SVD on the recurrent weights given by $\boldsymbol{W}_{rec}(r, s) = \boldsymbol{U\Sigma V}^T$. Furthermore, consider a perturbation $\boldsymbol{e}$ sampled uniformly at random from the set of norm-1 vectors. If we apply this perturbation to the hidden state $\boldsymbol{h}_t$, then to measure robustness we want to quantify the deviation between $\boldsymbol{W}_{rec}(r, s)\boldsymbol{h}_t$ and $\boldsymbol{W}_{rec}(r, s)(\boldsymbol{h}_t + \boldsymbol{e})$. This is given by $\boldsymbol{W}_{rec}(r, s)\boldsymbol{e} = \boldsymbol{U\Sigma V}^T\boldsymbol{e}$. Note that both $\boldsymbol{U}$ and $\boldsymbol{V}^T$ are unitary matrices and thus do not affect the magnitude of $\boldsymbol{e}$. This means that $\boldsymbol{\Sigma}$ captures the nature of the transformation on $\boldsymbol{e}$. The two relevant aspects of the transformation are its magnitude and direction. The spectral norm, the maximum singular value, measures the former (i.e. smaller spectral norm implies better robustness). To quantify the latter, we examine the rate at which the singular values decay. This provides a proxy for the effective number of directions $\boldsymbol{e}$ is expanded in (i.e. faster decay implies better robustness). For details on this argument, refer to A.5.2.

Now that we have motivated the spectral properties of $\boldsymbol{W}_{rec}(r, s)$, we next aim to understand how they change as a function of rank and sparsity. We consider two random initialization schemes for $\boldsymbol{W}_{rec}(r, s)$: Glorot uniform spectral initialization (GU-spec) and orthogonal spectral initialization (ortho-spec). To thoroughly motivate the proposed parameterization, we provide theoretical proof of the relationships between rank/sparsity and spectral radius/spectral norm for both initialization schemes where possible. For the cases we do not prove, we provide an empirical analysis instead.

**Theorem 1.** *Given recurrent weights with parametrization shown in Equation (2) and initialization scheme specified in parentheses, we prove the following:*

- *The spectral radius of $\boldsymbol{W}_{rec}(r, s)$ decreases as a function of $s$ (GU-spec)*

- *The spectral radius of $\boldsymbol{W}_{rec}(r, s)$ increases as a function of $r$ (ortho-spec)*

- *The spectral norm of $\boldsymbol{W}_{rec}(r, s)$ decreases as a function of $s$ (GU-spec)*

- *The spectral norm of $\boldsymbol{W}_{rec}(r, s)$ is constant as a function of $r$ (ortho-spec, GU-spec)*

For proofs of the above properties and an empirical analysis for the unproven cases (which follow the same trends as the proven cases), refer to A.6. And since our theoretical arguments do not readily extend to the full singular value spectrum, we provide an empirical analysis as a function of rank and sparsity for this as well (A.7) which shows that the rate of singular value spectrum decay increases as a function of rank and decreases as a function of sparsity.

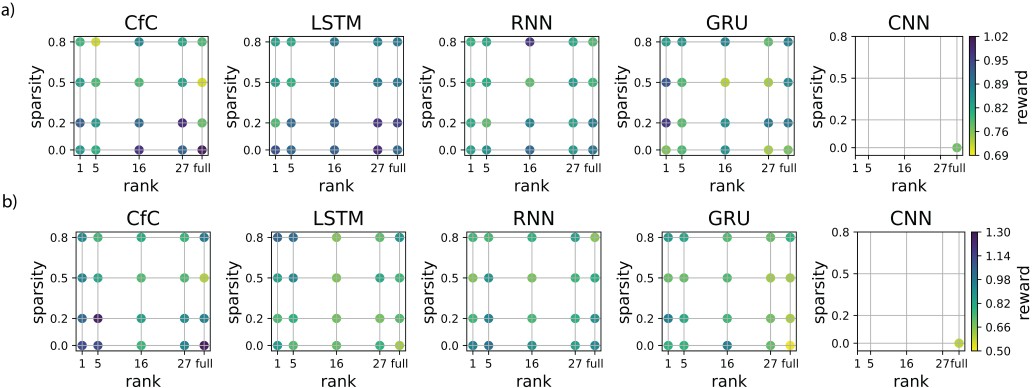

Figure 3: Online performance of recurrent networks under different ranks and sparsities in Seaquest environment. For offline performance, refer to A.9. a) In-distribution rewards in the online, closed-loop setting normalized by the rewards obtained by the expert in-distribution. b) Rewards averaged across 5 distribution shifts, normalized by the rewards obtained by the expert under distribution shift.

## 4  EXPERIMENTS

**Experimental setup.** We study the offline-online generalization gap of various recurrent architectures parameterized by low-rank, sparse connectivity under an imitation learning framework. In particular, we measure the performance of our models in the Arcade Learning Environment (Bellemare et al., 2012) and MuJoCo (Todorov et al., 2012). For ALEs, we run experiments in the Seaquest and Alien environments and for MuJoCo we explore the HalfCheetah environment. For each environment, we train a deep-Q network to generate expert trajectories that we then use to fit our recurrent networks in an offline setting. We evaluate the recurrent networks online, closed-loop when deployed into the environment, both in-distribution and under distribution shift. For more details on the experimental framework, refer to A.11.

**Models.** We examine the following recurrent architectures: RNNs, LSTMs, GRUs, CNNs and CfCs. RNNs, LSTMs (Hochreiter & Schmidhuber, 1997) and GRUs (Cho et al., 2014) refer to their canonical implementations. CNNs serve to address whether supervision along the recurrent dimension is even necessary. CfCs (Hasani et al., 2021) are a modified version of an RNN that admit the following functional form:

$$\boldsymbol{h}(t) = \sigma(F(\boldsymbol{h}_{t-1}, \boldsymbol{x}_t, \theta_F)) \odot G(\boldsymbol{h}_{t-1}, \boldsymbol{x}_t, \theta_G) + [1 - \sigma(F(\boldsymbol{h}_{t-1}, \boldsymbol{x}_t, \theta_f)] \odot H(\boldsymbol{h}_{t-1}, \boldsymbol{x}_t, \theta_H)$$

Here, $H$ and $G$ are vanilla RNNs and $F$ can be interpreted as an adaptive gating mechanism that interpolates between the state-space trajectories of $H$ and $G$ on a per-element basis in the hidden vector $\boldsymbol{h}(t)$ (refer to A.4 for details). Finally, recall in Section 3.1 that we proposed a low-rank, sparse parameterization of connectivity $\boldsymbol{W}_{rec}(r, s)$ for the recurrent weights in a vanilla RNN. We generalize this parameterization to the other recurrent architectures by simply parameterizing all recurrent weights in the network in the form of $\boldsymbol{W}_{rec}(r, s)$. For specifics, refer to A.11.

### 4.1  PERFORMANCE IN CLOSED-LOOP SETTINGS

We first explore the impact of our proposed connectivity parameterization for various ranks $r$ and sparsities $s$ in the Seaquest environment in the online, in-distribution setting. We observe that the best performing models in-distribution tend to be low-sparsity, high-rank CfCs and LSTMs (Figure 3a). GRUs, on the other hand, tend to perform poorly for most $(r, s)$ which aligns with previous work which also shows that GRUs are not particularly well-suited for closed-loop settings (Chahine et al., 2023). Due to the poor performance of GRUs relative to LSTMs, we will restrict most of our discussion and analysis to CfCs, LSTMs and RNNs. The most notable finding from the in-distribution results is observed in the high-sparsity models; in particular, we find that LSTMs tend to perform

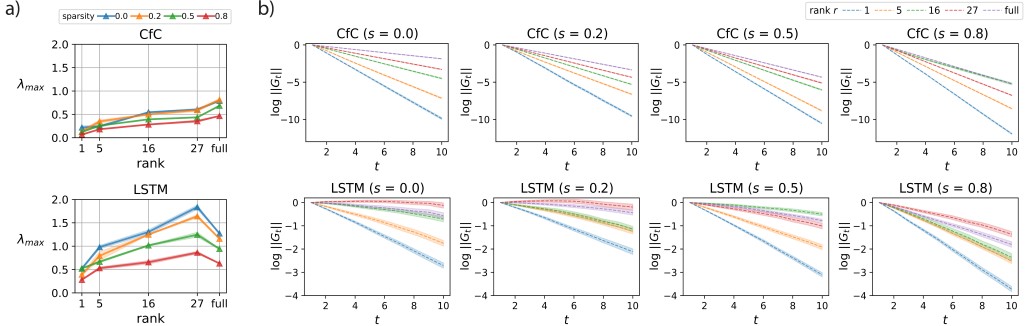

Figure 4: a) Spectral radius of recurrent weights $\boldsymbol{W}_{rec}(r, s)$ in trained CfC and LSTM networks across ranks and sparsities ($\pm 1$ SE). b) Frobenius norm of recurrent gradients $\boldsymbol{G}_t$ as a function of time ($\pm 1$ SE). Note that the norms are plotted in log space and translated to decay from 0 to enable easier comparison. For details on the computation, refer to A.11.

much better than CfCs at high sparsities. Analogously, RNNs, which in general tend to perform worse than CfCs, show similarly worse performance at high sparsities relative to LSTMs. We offer intuition for these findings in Section 4.2 where we explore the relationship between sparsity and the recurrent memory horizon of the network.

Under distribution shift, we find that the best performing models are low-rank, low-sparsity CfCs. More generally, across all recurrent architectures, we find that the low-rank models tend to perform on par with, and in many cases actually outperform, higher rank ones. This demonstrates that we can construct models with fewer parameters at initialization yet still learn more robust models. Interestingly, however, we find that the means through which we prune at initialization matters: in particular for high-sparsity models, we do not achieve the same robustness that we observe in low-rank ones. We will provide justification for the apparent disparity between pruning by increasing sparsity and pruning by decreasing rank in Section 4.3. For analogous results in the Alien and HalfCheetah environments, refer to A.9.

## 4.2 MODULATING RECURRENT MEMORY

Here, we aim to gain intuition for the results we observed in the online, in-distribution setting and in particular offer an explanation as to why the only effective form of pruning in-distribution was inducing sparsity in LSTMs. Recall in Section 3.1 we motivated the parameterization $\boldsymbol{W}_{rec}(r, s)$ with respect to three measures, one of which was spectral radius, a proxy for a model's attention span across time. In particular, we showed at initialization that the spectral radius of $\boldsymbol{W}_{rec}(r, s)$ decreases as a function of sparsity and increases as a function of rank. We first note that these trends persist after training (Figure 4a). In particular, in both CfCs and LSTMs, the low-rank and highly-sparse models tend to admit solutions with recurrent weights of low spectral radius.

Comparing CfCs and LSTMs, we note that CfCs tend to admit solutions with significantly lower spectral radii. This is perhaps not surprising considering that LSTMs explicitly promote a more consistent error flow over time (Jozefowicz et al., 2015) via their forget gate, and are thus more likely to attend to distant observations. So, assuming that the spectral radius is a reasonable proxy for the recurrent memory-horizon of a model, it appears that LSTMs tend to have longer-term memory than CfCs. However, note that the quantity we care about in practice is $\left\| \frac{\partial \boldsymbol{h}_k}{\partial \boldsymbol{h}_{k-1}} \right\| = \|\boldsymbol{J}_k\|$ as this is the true measure of gradient propagation backwards in time. Since $\|\boldsymbol{J}_k\|$ depends not only on the recurrent weights but also on the hidden state (for details, refer to A.4), it is possible that the disparate functional forms of CfCs and LSTMs mitigates the efficacy of spectral radius as a proxy for attention across time (for more details on this intuition, refer to A.5.1). Let us define $\boldsymbol{G}_t = \boldsymbol{J}_k \boldsymbol{J}_{k-1} \ldots \boldsymbol{J}_{k-t+1}$. Assuming that time $k$ represents the end of the time series, $\boldsymbol{G}_t$ represent the gradient backpropagated $t$ steps in time, which when analyzed as a function of $t$ provides a direct measure of how much importance a model assigns to inputs as a function of the time that has passed since it last observed them. In particular, we compute $\log \|\boldsymbol{G}_t\|$ as a function of $t$ and

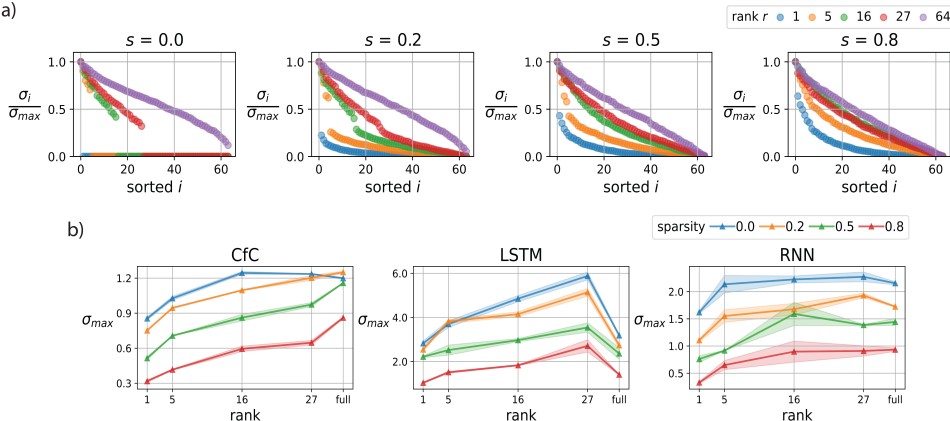

Figure 5: a) Decay rate of singular value spectrum of CfC recurrent weights as a proxy for robustness. b) Spectral norm of the recurrent weights as a proxy for robustness ($\pm 1$ SE).

find that gradients decay significantly faster across time in CfCs than LSTMs (Figure 4b). One, this demonstrates that spectral radius is an effective measure of recurrent memory across architectures in this task setting. Two, we observe that in their full-rank, fully-connected forms, LSTMs have a long recurrent memory-horizon, while CfCs do not. In the context of a closed-loop task, we know that it is beneficial to limit the network's temporal attention span (Lechner et al., 2020). Since an LSTM's recurrent gradient in its baseline form does not decrease particularly fast across time, inducing sparsity at initialization pushes the network into the vanishing gradient regime, reducing its attention across time. This result provides support as to why LSTMs are quite effective at navigating environments in-distribution in spite of high-sparsity in their recurrent weights (Figure 3a). In contrast, CfCs in their baseline form already lie in the vanishing gradient regime: intuitively, inducing sparsity is less effective when the baseline network inherently possesses an affinity to selectively attend to past observations. This aligns with an analogous trend observed in RNNs which are networks that, like CfCs, are amenable to learning a vanishing gradient (Figure 19).

By analyzing the spectral radius of $\boldsymbol{W}_{rec}(r, s)$ as well as the recurrent gradients $\boldsymbol{G}_t$, we have characterized sparsity in its ability to modulate the network's recurrent memory-horizon. However, note that we also observe that low-rank LSTMs tend to admit solutions with lower spectral radius as well (Figure 4b). Yet, low-rank LSTMs tend to perform worse in-distribution (Figure 3a) than their sparse counterparts. To understand why there is a disparity between reducing rank and increasing sparsity, we turn to an analysis of the singular values, which are best motivated in the distribution shift setting.

### 4.3 ROBUSTNESS UNDER DISTRIBUTION SHIFT

In Section 4.1, we found that in the distribution shift setting, pruning by reducing rank improved performance (most notably in CfCs) whereas pruning by increasing sparsity did not. This is in stark juxtaposition to the in-distribution trends in which inducing sparsity in LSTMs was the only form of pruning that did not worsen performance (Section 4.2). Here, we attempt to interpret these results by understanding both why CfCs appear to be the most robust model type and why pruning by reducing rank is distinct from pruning by increasing sparsity.

Recall in Section 3.1 we reduced the robustness of the hidden state under perturbation to two measures: the spectral norm and the decay of the singular value spectrum of $\boldsymbol{W}_{rec}(r, s)$ (for intuition, refer to A.5.2). Regarding the former, across models we observe that CfCs have lower spectral norms than both RNNs and LSTMs (Figure 5b, Figure 23). While this offers intuition as to the robustness of CfCs, it is not sufficient as a standalone measure. This is because in practice, distribution shifts are applied to the input $\boldsymbol{x}_t$, which in turn corrupts the hidden state $\boldsymbol{h}_{t+1}$. Thus, while it remains important to analyze the spectral norm of the recurrent weights, it is also pertinent to analyze the spectral norm of the input weights. In doing so, we find that while there exist marginal differences across architectures, the input spectral norm does not vary to the extent we observe in the recurrent

spectral norm (Figure 20b). This, along with the fact that CfCs learn significantly lower spectral norms in their recurrent weights, offers intuition as to the heightened robustness of CfCs under distribution shift. In addition, the (albeit loose) relationship between weights with lower spectral norms corresponding to networks with lower Lipschitz constants provides even further support as to why CfCs tend to express more robust functions (elaborated upon in A.5.2).

Next, we address the observed disparity between sparse and low-rank recurrent connectivity. Across ranks and sparsities, the spectral norm decreases as a function of increasing sparsity and decreasing rank (Figure 5b), aligning with the trends induced at initialization. In Section 4.2, we similarly showed that spectral radius decreases as a function of increasing sparsity and decreasing rank in the trained networks. Thus, we cannot explain the apparent disparity between low-rank and sparse connectivity via these measures, and instead turn to the decay of the singular values. Again, aligning with the prior induced at initialization, we find that the decay of the singular values increases as a function of increasing sparsity but decreases as a function of decreasing rank (Figure 5a). And this is precisely where the two methods of pruning differ: since sparsity reduces the rate of spectral decay, the resulting transformation (induced by $\boldsymbol{W}_{rec}$) on a perturbed version of the hidden state vector expands in more directions (i.e. increasing the effect of the perturbation); lowering the rank does the opposite. We can make this notion more concrete by analyzing the dimensionality of the state-space trajectories of each model. Note that the trajectory induced by $\boldsymbol{h}_t$ can be decomposed into two parts: the recurrently-driven portion $\boldsymbol{W}_{rec}\boldsymbol{h}_{t-1}$ and the input-driven portion $\boldsymbol{W}_{inp}\boldsymbol{x}_t$. Under this formulation, we can consider the full state-space trajectory $\boldsymbol{h}_t$ to be driven by $\boldsymbol{W}_{full} = [\boldsymbol{W}_{rec}\ \boldsymbol{W}_{inp}]$.

To measure the complexity of these state-space trajectories, we ran PCA on them to measure their effective dimensionalities. We find that with decreasing rank, the dimensionality of the recurrently-driven trajectories decreases, whereas with increasing sparsity it increases (Figure 6a, Figure 6b). Hence, our intuition is as follows: since the activity along the recurrent axis is constrained to lie in the subspace spanned by the vectors comprising $\boldsymbol{W}_{rec}(r, s)$, the recurrent dynamics of low-rank recurrent networks are lower-dimensional and hence simpler. We can imagine that in the in-distribution setting, it is not desirable to arbitrarily constrain the net-

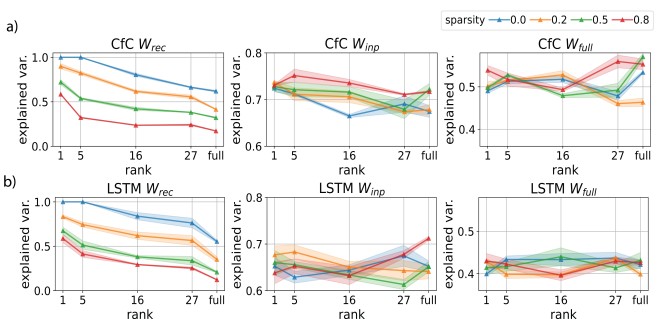

Figure 6: Effective dimensionalities of the recurrent, input and full state-space trajectories collected during online testing, in-distribution, as measured by the explained variance of the top 5 principal components. a) CfC state-space dynamics ($\pm 1$ SE). b) LSTM state-space dynamics ($\pm 1$ SE).

work's capacity to learn recurrently. This provides justification as to why, in spite of the low-rank LSTMs also having a shorter recurrent memory-horizon, inducing sparsity into LSTMs does not worsen the in-distribution performance of the network like constraining rank does (Figure 3a). In contrast, recall that under distribution shift, we observed the opposite: inducing sparsity at initialization was detrimental whereas constraining the network to be low-rank improved robustness. Constraining the recurrently-driven portion of the state-space is one means of reducing the network's variability in the presence of input perturbations. Supervision along the recurrent axis is pivotal when the agent is faced with environmental occlusions, as it needs to lean on some notion of the past to make a decision in the present. By making the recurrent state more robust, we are better able to generalize under distribution shift.

We make one final note across the model axis to further justify why CfCs appear to be inherently more robust than LSTMs and RNNs. In particular, in both the input-driven and full state-space trajectories (each of which are little affected by changes in the recurrent connectivity), we find that for all pairs $(r, s)$, the dimensionality of LSTM trajectories is much higher than that of CfC trajectories, offering additional justification for the robustness observed in CfCs. But what is perhaps more surprising is that RNNs, despite their simpler functional form, also posses higher-dimensional state-space dynamics than CfCs (Figure 18). Since RNNs and CfCs differ only with respect to the time constant network $F$, we explore this gating mechanism further in A.2.

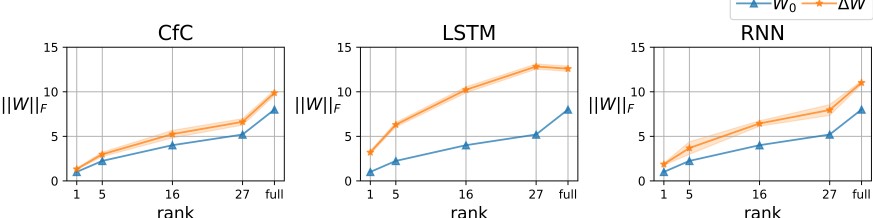

Figure 7: We individually consider the Frobenius norm of the weights at initialization $\boldsymbol{W}_0$ and the change in the weights after training $\Delta \boldsymbol{W}$. Note that the models shown here have no sparsity.

### 4.4 EXPLORING THE TASK DIMENSION GAP

We now have demonstrated two things: one is the efficacy of the proposed connectivity parameterization in units of its interpretability as a modulator of network dynamics. Second is the impact connectivity has on each of the networks we examined: in particular, by disentangling the effects of rank and sparsity, we showed why LSTMs are more amenable to sparse connectivity in-distribution whereas CfCs tend to show the most promise with low-rank connectivity under distribution shift. Here, we further our intuition on the performance of CfCs under distribution shift by understanding why they tend to be the most amenable architecture to low-rank recurrent connectivity.

As we have noted, the efficacy of this parameterization rests upon the ability of the network to adhere to the prior throughout training. And recall that we found that the spectral norm of the recurrent weights in CfCs remained much closer to 1 in the full-rank, fully-connected network than in either LSTMs or RNNs (Figure 5). To better understand this, we borrow from an analysis conducted by Schuessler et al. (2020) in which they decomposed the recurrent weights as follows: $\boldsymbol{W}_{rec} = \boldsymbol{W}_0 + \Delta \boldsymbol{W}$ where $\boldsymbol{W}_0$ denotes the weights at initialization and $\Delta \boldsymbol{W}$ denotes the change in the weights after training. In their setting, the purpose of the decomposition was to demonstrate that in a set of simple tasks, the Frobenius norm of the weights at convergence $||\boldsymbol{W}_{rec}||_F$ is dominated by the norm of the weights at initialization $||\boldsymbol{W}_0||_F$. In spite of the recurrent connectivity they used being full-rank, they found that the changes in the weights learned during training were in fact low-rank as measured by $||\Delta \boldsymbol{W}||_F$. In our analysis, we find the opposite: $||\boldsymbol{W}_0||_F$ certainly does not dominate the norm of the final weights and is in fact lower than $||\Delta \boldsymbol{W}||_F$ (Figure 7). This reinforces the notion of task dimension put forth by Schuessler et al. (2020) which describes the rank of the training-induced connectivity changes as a function of the task the network is trained on. In particular, in their work, they showed that the task dimension of the simple tasks they examined was low and hence a network with unconstrained, full-rank connectivity learned low-rank changes. In contrast, in our setting we consider a significantly more complex task domain which incites higher-rank changes in the recurrent connectivity. This brings forth the notion of a task dimension gap between the offline, open-loop and online, closed-loop settings: namely, the networks we examined are trained offline without being exposed to distribution shifts and hence learn higher-rank changes in their connectivity. In contrast, as we have shown, succeeding in the closed-loop setting under distribution shift means learning lower-rank dynamics. Thus, networks that are able to abide to our low-rank prior and avoid learning high-rank changes in connectivity are better at generalizing under distribution shift. This is precisely where CfCs supersede LSTMs and to some extent RNNs as well. We find that despite each network starting at the same $||\boldsymbol{W}_0||_F$, $||\Delta \boldsymbol{W}||_F$ is lowest in CfCs.

## 5 CONCLUSION

In this work, we investigated the use of a low-rank, sparse parameterization of recurrent connectivity in various architectures as a means of improving model robustness in closed-loop environments. We showed that this type of connectivity was most amenable to CfCs and also showed promise in more canonical networks like LSTMs and RNNs. Furthermore, we demonstrated the interpretability of this prior by analyzing the network dynamics it induces as a function of both rank and sparsity. Our results represent an application in pruning recurrent networks at initialization to improve performance under distribution shift.

## ACKNOWLEDGMENTS

Research was sponsored by the United States Air Force Research Laboratory and the Department of the Air Force Artificial Intelligence Accelerator and was accomplished under Cooperative Agreement Number FA8750- 19-2-1000. The views and conclusions contained in this document are those of the authors and should not be interpreted as representing the official policies, either expressed or implied, of the Department of the Air Force or the U.S. Government. The U.S. Government is authorized to reproduce and distribute reprints for Government purposes notwithstanding any copyright notation herein.

## REPRODUCIBILITY STATEMENT

To ensure the reproducibility of our work, we extensively detail the experimental setup in the appendix as well as provide information regarding the analyses we conducted. The bulk of these details can be found in appendix A.11 which describes how we constructed the dataset, how the models were trained (including details on hyperparameters), the evaluation metrics for the models and the intuition/implementation regarding the analyses that we performed on the trained models. A key portion of our results is driven by an initialization scheme we proposed that deviates from default initializers given in existing open-source implementations. We thoroughly describe how and why our proposed initialization differs in appendix A.10, so the reader can leverage it to reproduce our results. Regarding our theoretical work, we provide proofs for the claims we made in the main portion of the paper which can be found in appendix A.6. In that section, we clearly delineate the cases in which we were unable to prove certain claims and had to resort to an empirical analysis instead.

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

# A APPENDIX

## A.1 DIMENSIONALITY ANALYSIS UNDER DISTRIBUTION SHIFT

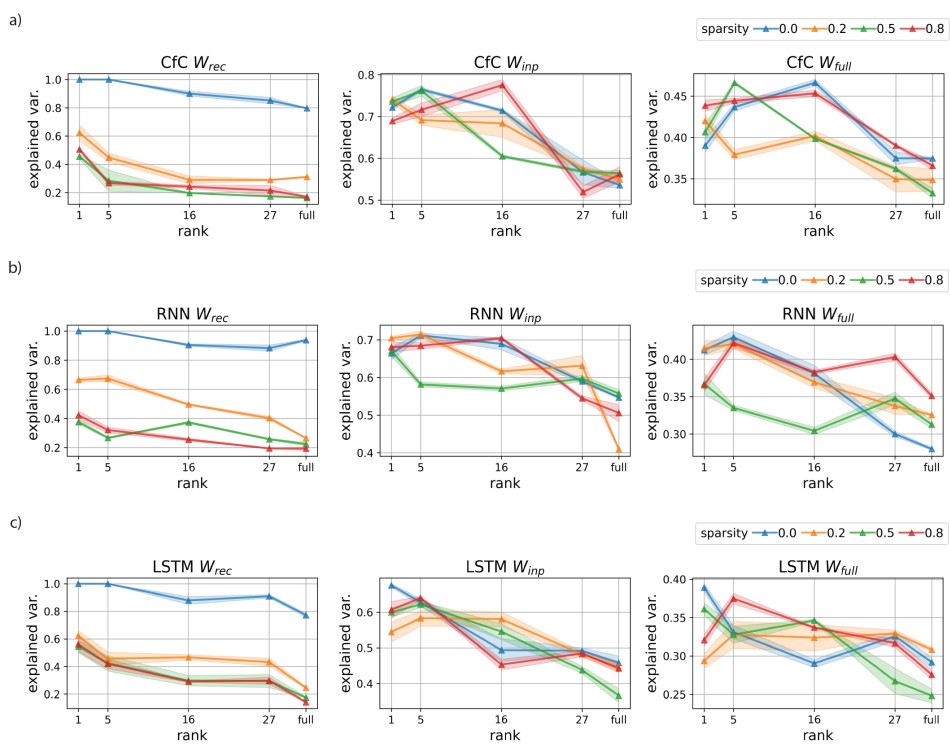

Figure 8: Effective dimensionalities of the recurrent, input and full state-space trajectories collected during online testing, under the noise distribution shift (for details, refer to A.11), as measured by the explained variance of the top 5 principal components. a) CfC state-space dynamics ($\pm 1$ SE). b) RNN state-space dynamics ($\pm 1$ SE). c) LSTM state-space dynamics ($\pm 1$ SE).

Recall in Figure 6, we computed the effective dimensionalities of the recurrent, input and full state-space trajectories across models, ranks and sparsities collected during an online, closed-loop simulations of the agents in an in-distribution setting. In doing so, we found that the input and full state space dimensionalities were not particularly affected by changes in the recurrent rank or sparsity. Here, we will show that the same does not hold under distribution shift and understand why this is a desirable property of the parameterization of recurrent connectivity.

In Figure 8, we find that in CfCs, RNNs and LSTMs, the effective dimensionalities of the input and full state spaces increases with the rank of the recurrent weights. This is interesting as it demonstrates the ability of the parameterization to modulate state-space dynamics outside of the recurrently-driven subspace of activity. But it also begs the question as to why we do not observe this effect in the in-distribution setting.

The primary distinction is that under distribution shift, we can imagine that adding noise to the inputs causes the input state-space trajectory to evolve in random directions, raising its dimensionality. Having a robust recurrent state presumably allows for the filtration of some of this noise, which reduces the effect it has on future model inputs. Furthermore, since we examine these trajectories in the context of a closed loop system, the input is itself a function of the previous hidden state. By making this function more robust via our low-rank prior, this in turn reduces the dimensionality of the input state-space trajectory (and by proxy the full state-space trajectory as well).

One final note is that while we observe this trend as a function of rank, we do not observe it as a function of sparsity. In particular, modulations in the dimensionality of the recurrent state-space via changes in the sparsity of the recurrent weights do not appear to impact the dimensionality of

input or full state-spaces (Figure 8). While this certainly requires further exploration, one possible hypothesis is that this is caused by the counteracting effects of sparsity on robustness. Namely, recall that sparsity both reduces the spectral radius which shortens the temporal attention span of the network, making the model more robust across time. However, it also reduces the decay rate of the singular value spectrum, making it less robust at any given point in time. These two effects potentially offset one another and prevent the sparsity in the recurrent weights from modulating the dimensionality of the input trajectory.

## A.2 Time constant analysis

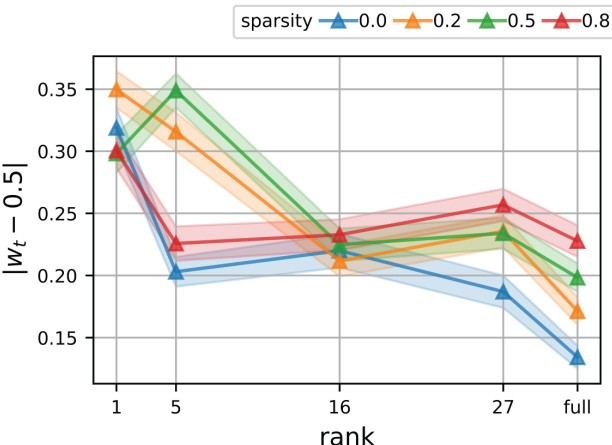

Figure 9: Average absolute deviation from $0.5$ of time constant vectors in trained CfC models.

In Section 4.3, we demonstrated that on average, CfCs both learn lower spectral norms in their recurrent weights and express lower dimensional state-space trajectories than RNNs. Recall, the functional form of a CfC differs from an RNN via the time constant network $F(\boldsymbol{h}_{t-1}, \boldsymbol{x}_t)$ which learns a vector of weights used to interpolate between $G$ and $H$. Note that if $F(\boldsymbol{h}_{t-1}, \boldsymbol{x}_t) = [1, 1, \ldots, 1]$, the network reduces to that of an RNN by placing all of its weight on a single trajectory. Because the time-constant learned by $F$ is a function of the input, the network can dynamically adapt its interpolation weights under distribution shift by changing how much it weighs the trajectory induced by $G$ versus the trajectory induced by $H$. While rationale beyond the effects of the time-constant module on the overall network dynamics warrants further exploration, the notion of gating mechanisms improving performance in neural networks is not a new one; recent work done in convolutional language models shows similar findings (Poli et al., 2023).

In any case, to better understand the time constant network, here we provide an analysis on what role modulating recurrent sparsity and rank has on the learned interpolation weights. In particular, let $\boldsymbol{w}_t$ denote the interpolation weights that are the output of $F$ at time $t$. We compute

$$|\boldsymbol{w}_t - 0.5| = \frac{\sum_{t=1}^{T} \sum_{i=1}^{h} |w_{ti} - 0.5|}{hT}$$

which measures the average deviation of the weights from $0.5$. We consider this metric as a means of understanding how much the interpolation between $G$ and $H$ tends to rely upon only one of the trajectories (as we observe in the case of an RNN) versus evenly combining them (as would be the case if $F$ learns to output $[\frac{1}{2}, \frac{1}{2}, \ldots, \frac{1}{2}]$). We find that as a function of increasing rank, the deviation of the time constant from $0.5$ decreases (Figure 9). This means that low-rank CfCs tend to place more of their weight on one trajectory than the other which demonstrates the ability of the time constant network to align with the simpler recurrent dynamics promoted in the low-rank setting.

## A.3 Task dimension continued

Here, we provide some additional commentary on our discussion of the task dimension gap in Section 4.4.

At a high level, the notion of a task dimension gap which characterizes the disconnect between agents learning passively in the offline setting and learning actively in the online setting warrants further exploration and formalization. Furthermore, we note that, the notion of task dimension put forth by Schuessler et al. (2020) itself is more aptly decomposed into recurrent task dimension and input task dimension. In particular, as we have shown, changes in recurrent connectivity are not necessarily synonymous with changes in input connectivity (Figure 6). A given task may necessitate learning connectivities of different ranks along the input axis and recurrent axis: disentangling the two is a necessity in understanding the dynamics of recurrent neural networks. Another point to note is that recurrent task dimension is distinct from attention span along the recurrent axis, which can also be considered a function of the task. We showed that while rank and sparsity both reduce temporal attention span, they have opposing effects on the dimensionality of recurrently-driven activity. Noting this distinction is pivotal in properly characterizing recurrent dynamics.

## A.4   DERIVING RECURRENT GRADIENTS

In this section, we provide details on the derivation for $J_t = \frac{\partial h_t}{\partial h_{t-1}}$ in RNNs, LSTMs and CfCs. $J_t$ is the Jacobian of the hidden-state dynamics which captures information about the rate at which the gradient is propagated across time. We motivate this further when we discuss the relationship between $J_t$ and the memory-horizon of a recurrent network in Section A.5.

### A.4.1   RNN

Recall that the functional form of an RNN is given by

$$h_t = \tanh(W_h h_{t-1} + W_i x_t + b)$$

where $W_h$ denotes the recurrent weights, $W_i$ denotes the input weights and $b$ denotes the bias. Then, if we let $\odot$ denote row-wise multiplication in the case of a vector and matrix, then we have that

$$\frac{\partial h_t}{\partial h_{t-1}} = (1 - \tanh^2(h_{t-1})) \odot W_h$$

### A.4.2   LSTM

The functional form of an LSTM is given by

$$
\begin{aligned}
i_t &= \sigma(W_{ii} x_t + W_{hi} h_{t-1} + b_i) \\
f_t &= \sigma(W_{if} x_t + W_{hf} h_{t-1} + b_f) \\
o_t &= \sigma(W_{io} x_t + W_{ho} h_{t-1} + b_o) \\
g_t &= \tanh(W_{ic} x_t + W_{hc} h_{t-1} + b_c) \\
c_t &= f_t \odot c_{t-1} + i_t \odot g_t \\
h_t &= o_t \odot \tanh(c_t)
\end{aligned}
$$

Following the notation from Vogt et al. (2020), if we let

$$y_* = W_{i*} x_t + W_{h*} h_{t-1} + b_*$$

where $*$ is determined by the gate and $\odot$ denotes elementwise multiplication in the case of two vectors and row-wise multiplication in the case of a vector and matrix, then we have that

$$\frac{\partial \boldsymbol{f}_t}{\partial \boldsymbol{h}_{t-1}} = [\sigma(\boldsymbol{y}_f) \odot (1 - \sigma(\boldsymbol{y}_f))]^T \odot \boldsymbol{W}_{hf}$$

$$\frac{\partial \boldsymbol{i}_t}{\partial \boldsymbol{h}_{t-1}} = [\sigma(\boldsymbol{y}_i) \odot (1 - \sigma(\boldsymbol{y}_i))]^T \odot \boldsymbol{W}_{hi}$$

$$\frac{\partial \boldsymbol{o}_t}{\partial \boldsymbol{h}_{t-1}} = [\sigma(\boldsymbol{y}_o) \odot (1 - \sigma(\boldsymbol{y}_o))]^T \odot \boldsymbol{W}_{ho}$$

$$\frac{\partial \boldsymbol{g}_t}{\partial \boldsymbol{h}_{t-1}} = [(1 - \tanh^2(\boldsymbol{y}_c))]^T \odot \boldsymbol{W}_{hc}$$

$$\frac{\partial \boldsymbol{c}_t}{\partial \boldsymbol{h}_{t-1}} = \frac{\partial \boldsymbol{f}_t}{\partial \boldsymbol{h}_{t-1}} \odot \boldsymbol{c}_{t-1} + \frac{\partial \boldsymbol{i}_t}{\partial \boldsymbol{h}_{t-1}} \odot \tanh(\boldsymbol{y}_c) + \boldsymbol{i}_t \odot (1 - \tanh^2(\boldsymbol{y}_c)) \odot \boldsymbol{W}_{hc} + \boldsymbol{f}_t$$

$$\frac{\partial \boldsymbol{h}_t}{\partial \boldsymbol{h}_{t-1}} = \frac{\partial \boldsymbol{o}_t}{\partial \boldsymbol{h}_{t-1}} \odot \tanh(\boldsymbol{c}_t) + \boldsymbol{o}_t \odot (1 - \tanh^2(\boldsymbol{c}_t)) \odot \frac{\partial \boldsymbol{c}_t}{\partial \boldsymbol{h}_{t-1}}$$

### A.4.3 CFC

The functional form of a CfC is given by

$$\boldsymbol{h}(t) = \sigma(F(\boldsymbol{h}_{t-1}, \boldsymbol{x}_t, \theta_F)) \odot G(\boldsymbol{h}_{t-1}, \boldsymbol{x}_t, \theta_G) + [1 - \sigma(F(\boldsymbol{h}_{t-1}, \boldsymbol{x}_t, \theta_f)] \odot H(\boldsymbol{h}_{t-1}, \boldsymbol{x}_t, \theta_H)$$

where $\theta_F, \theta_G, \theta_H$ refer to the parameters in each module as given by the following equations:

$$F(\boldsymbol{h}_{t-1}, \boldsymbol{x}_t, \theta_F) = \boldsymbol{W}_{if}\boldsymbol{x}_t + \boldsymbol{W}_{hf}\boldsymbol{h}_{t-1} + \boldsymbol{b}_f$$
$$G(\boldsymbol{h}_{t-1}, \boldsymbol{x}_t, \theta_G) = \tanh(\boldsymbol{W}_{ig}\boldsymbol{x}_t + \boldsymbol{W}_{hg}\boldsymbol{h}_{t-1} + \boldsymbol{b}_g)$$
$$H(\boldsymbol{h}_{t-1}, \boldsymbol{x}_t, \theta_H) = \tanh(\boldsymbol{W}_{ih}\boldsymbol{x}_t + \boldsymbol{W}_{hh}\boldsymbol{h}_{t-1} + \boldsymbol{b}_h)$$

We note that in the work that originally proposed the CfC architecture (Hasani et al., 2021), $F$ is referred to as a liquid time constant network due to its motivation as a time constant in a dynamical system. In general, time constants are more thoroughly motivated in the continuous-time setting in which a neural network is used to model the derivative of the hidden state as opposed to the hidden state itself (Chen et al., 2018). In that setting, a time constant represents a parameter that characterizes the speed and coupling sensitivity of an ODE that models a system. However, our work exists in the discrete-time setting in which we can no longer interpret the time constant as a parameter of a continuous-time system. Instead, we interpret $F$ as an adaptive gating mechanism, as discussed in Section 4.

If we let
$$\boldsymbol{y}_* = \boldsymbol{W}_{i*}\boldsymbol{x}_t + \boldsymbol{W}_{h*}\boldsymbol{h}_{t-1} + \boldsymbol{b}_*$$
where $*$ is determined by the gate, then it follows that

$$\frac{\partial \boldsymbol{h}_t}{\partial \boldsymbol{h}_{t-1}} = \sigma(\boldsymbol{y}_f) \odot (1 - \tanh^2(\boldsymbol{y}_g))\boldsymbol{W}_{hg} + \tanh(\boldsymbol{y}_g) \odot [\sigma(\boldsymbol{y}_f) \odot (1 - \sigma(\boldsymbol{y}_f)) \odot \boldsymbol{W}_{hf}]$$

$$+ [1 - \sigma(\boldsymbol{y}_f)] \odot (1 - \tanh^2(\boldsymbol{y}_h))\boldsymbol{W}_{hh} + \tanh(\boldsymbol{y}_h) \odot [1 - (\sigma(\boldsymbol{y}_f) \odot (1 - \sigma(\boldsymbol{y}_f)) \odot \boldsymbol{W}_{hf})]$$

### A.5 MOTIVATING SPECTRAL ANALYSES OF RECURRENT MODELS

In this section, we motivate the analyses employed in this paper in order to analyze the dynamics of the various recurrent networks we examined. In particular, we will motivate the low-rank, sparse parameterization of $\boldsymbol{W}_{rec}$ from the perspective of modulating the spectral radius and spectral norm (and more generally the eigenspectrum and singular value spectrum) of the recurrent weights at initialization and then extend this line of reasoning to the analyses performed on the trained models. For specifics on the implementation details and computation performed for these analyses, refer to A.11.5.

A.5.1 RECURRENT MEMORY HORIZON

We leverage our parameterization of $\boldsymbol{W}_{rec}(r, s)$ as a function of rank and sparsity in order to modulate the spectral radius, spectral norm and singular value spectrum of the recurrent weights.

Here, we argue that the spectral radius of $\boldsymbol{W}_{rec}(r, s)$ is a pertinent measure for the memory horizon of the network across time. We will demonstrate why in the context of an RNN as the computations are most tractable under this functional form. In particular, in an RNN, the recurrent gradient in $\boldsymbol{J}_t = \frac{\partial \boldsymbol{h}_t}{\partial \boldsymbol{h}_{t-1}}$ reflects how much the network's hidden state is updated based on information from the past. A higher recurrent gradient suggests the network is paying more attention to distant inputs during training, while a lower recurrent gradient implies less reliance on such distant information.

The relevant quantity in backpropagation through time in an RNN is

$$\frac{\partial \boldsymbol{h}_t}{\partial \boldsymbol{h}_k} = \prod_{t \geq i \geq k} \frac{\partial \boldsymbol{h}_i}{\partial \boldsymbol{h}_{i-1}} = \prod_{t \geq i \geq k} (1 - \tanh^2(\boldsymbol{h}_{t-1})) \odot \boldsymbol{W}_h$$

(refer to A.4 for details on the derivation of RNN recurrent gradient). We can re-express the element-wise product in the expression for $\frac{\partial \boldsymbol{h}_i}{\partial \boldsymbol{h}_{i-1}}$ as the product of two matrices as follows:

$$diag[(1 - \tanh^2(\boldsymbol{h}_{t-1}))]\boldsymbol{W}_h$$

where the $diag(v)$ operator constructs a diagonal matrix where the elements of $v$ are placed along the diagonal. Then, we have that

$$\left\|\frac{\partial \boldsymbol{h}_i}{\partial \boldsymbol{h}_{i-1}}\right\| \leq \left\|diag[(1 - \tanh^2(\boldsymbol{h}_{t-1}))]\right\| \|\boldsymbol{W}_h\|$$

by the sub-multiplicativity of a matrix norm. As a function of rank and sparsity, if we assume that $\left\|diag[(1 - \tanh^2(\boldsymbol{h}_{t-1}))]\right\|$ is reasonably unaffected by the changes in $\boldsymbol{W}_{rec}(r, s)$ *at initialization*, then variation in the bound arises only from $\|\boldsymbol{W}_h\|$.

Drawing from the analysis presented in Pascanu et al. (2012), if we assume that the relevant variation in the bound as a function of rank and sparsity comes only from $\|\boldsymbol{W}_h\|$ and that $\boldsymbol{W}_h = \boldsymbol{P}\boldsymbol{D}\boldsymbol{P}^{-1}$ is diagonalizable, then we approximately have that

$$\left\|\frac{\partial \boldsymbol{h}_t}{\partial \boldsymbol{h}_k}\right\| \leq \left\|\prod_{t \geq i \geq k} diag[(1 - \tanh^2(\boldsymbol{h}_{i-1}))]\boldsymbol{W}_h\right\| \approx \left\|(\boldsymbol{W}_h)^{t-k}\right\| = \left\|\boldsymbol{P}\boldsymbol{D}^{t-k}\boldsymbol{P}^{-1}\right\|$$

where $\boldsymbol{P}$ denotes a matrix of eigenvectors and $\boldsymbol{D}$ denotes a diagonal matrix with the eigenvalues on the diagonal. It follows that for sufficiently large $\ell = t - k$, $\left\|\boldsymbol{P}\boldsymbol{D}^{t-k}\boldsymbol{P}^{-1}\right\|$ is dominated by the eigenvalue of leading magnitude. One can argue this more formally using the power iteration method, details of which can be found in Pascanu et al. (2012). We further note that this notion of modifying the spectral radius in order to modulate attention across the time in recurrent networks is not a new one. One prominent example can be found in echo state networks which like our parameterization induces sparsity in the recurrent weights in order to control how much attention the model pays to distant inputs.

Extending this mathematical argument to LSTMs and CfCs is less straightforward given the more intricate gating mechanisms present in the functional forms of each model. While it is reasonable to hypothesize that decreasing the spectral radius in these architectures will result in a faster decay of gradients across time, it is unclear how fast/slow this decay is relative to the analysis presented above for an RNN. Furthermore, the assumption made in the argument above for gradient decay in RNNs rested upon ignoring the portion of the gradient influenced by the hidden state: $\left\|diag[(1 - \tanh^2(\boldsymbol{h}_{t-1}))]\right\|$. This assumption becomes less reasonable after training as the network could potentially learn hidden-state vectors of small magnitude, causing the gradients to decay even faster. And this assumption is *even less reasonable* in LSTMs and CfCs after training, again due to the nuanced gating present in each architecture.

Thus, while we still examine the spectral radii of each architecture, we also conduct a more nuanced recurrent memory analysis on all the architectures by computing the norm of the recurrent gradients

backpropagated through time. In particular, if we let $\boldsymbol{J}_t = \frac{\partial \boldsymbol{h}_t}{\partial \boldsymbol{h}_{t-1}}$ denote the recurrent Jacobian, then we can take the cumulative product of the Jacobians as follows:

$$[\boldsymbol{J}_t, \boldsymbol{J}_t\boldsymbol{J}_{t-1}, \cdots, \boldsymbol{J}_t\boldsymbol{J}_{t-1}\dots\boldsymbol{J}_2, \boldsymbol{J}_t\boldsymbol{J}_{t-1}\dots\boldsymbol{J}_2\boldsymbol{J}_1]$$

Taking the norm of each of the matrices in the list above gives us a concrete measure of the extent to which a given model attends to its past observations as a function of time. This enables us to evaluate the effect of the recurrent weight's spectral radius on the memory-horizon of a given architecture which allows us to make comparisons not only within a given architecture across ranks and sparsities, but also across the different recurrent architectures we analyze (i.e. RNN vs LSTM vs CfC).

### A.5.2 ROBUSTNESS UNDER DISTRIBUTION SHIFT

In the last section, we motivated the parameterization of $\boldsymbol{W}_{rec}(r, s)$ from the perspective of the spectral radius and the implications it has on the attention profile of the network across time. In this section, we will motivate the parameterization instead from the perspective of the singular value spectrum of $\boldsymbol{W}_{rec}(r, s)$ and understand the implications it has on the robustness of the network under distribution shift.

Before motivating the analysis of the singular value spectrum, we first clarify the nature of the distribution shifts in the context of this work. Here, distribution shifts are applied to the input image, which is then fed through a set of convolutional layers before entering the recurrent portion of the network (A.11). The perturbed input denoted by $\boldsymbol{x}_t^*$ then corrupts the hidden state $\boldsymbol{h}_t^*$ via the update rule for the hidden state specified by the recurrent model. To simplify our robustness analysis, we do not consider the convolutional layers and instead restrict the scope of our analysis to the input weights and recurrent weights of the recurrent network. Since both $\boldsymbol{x}_t$ and $\boldsymbol{h}_t$ are affected by distribution shift, in practice we care about the robustness induced by both $\boldsymbol{W}_{inp}$ and $\boldsymbol{W}_{rec}$. However, recall that our parameterization only constructs $\boldsymbol{W}_{rec}(r, s)$ as a function of rank and sparsity, while maintaining the structure of $\boldsymbol{W}_{inp}$ as full-rank and fully-connected. We note that we did try extending the parameterization as a function of rank and sparsity to $\boldsymbol{W}_{inp}$, but found that doing so was quite detrimental to performance (results not shown). Thus, at initialization, we only modulate the robustness across the recurrent axis via the recurrent weights. Nonetheless, we still examine the spectral properties of the input weights after training to understand whether modulating the rank and sparsity of the recurrent weights implicitly affects the input weights during learning.

Now, let us more formally understand why modulating the spectral properties of the weights has an impact on robustness. Consider a perturbation $\boldsymbol{e}$ applied to the hidden state $\boldsymbol{h}_t$ such that $\|\boldsymbol{e}\| = 1$ (in practice recall that the perturbation is actually applied to $\boldsymbol{x}_t$ which later corrupts $\boldsymbol{h}_t$, but we apply the perturbation directly to $\boldsymbol{h}_t$ to simplify our argument). Under distribution shift, we care about our robustness against all possible perturbations since we can imagine $\boldsymbol{e}$ being sampled from some arbitrary distribution over unit vectors. In particular, we want to understand how $\boldsymbol{W}_{rec}\boldsymbol{h}_t$ differs from $\boldsymbol{W}_{rec}(\boldsymbol{h}_t + \boldsymbol{e})$. To motivate the importance of the singular value spectrum, consider an SVD on $\boldsymbol{W}_{rec}$ as follows:

$$\boldsymbol{W}_{rec}(\boldsymbol{h}_t + \boldsymbol{e}) = \boldsymbol{U}\boldsymbol{\Sigma}\boldsymbol{V}^T(\boldsymbol{h}_t + \boldsymbol{e}) = \boldsymbol{U}\boldsymbol{\Sigma}\boldsymbol{V}^T\boldsymbol{h}_t + \boldsymbol{U}\boldsymbol{\Sigma}\boldsymbol{V}^T\boldsymbol{e}$$

Note that the quantity we care about for measuring robustness is $\boldsymbol{U}\boldsymbol{\Sigma}\boldsymbol{V}^T\boldsymbol{e}$. Since $\boldsymbol{V}^T$ is a unitary matrix, $\boldsymbol{V}^T\boldsymbol{e} = \boldsymbol{e}^*$ has the same magnitude as $\boldsymbol{e}$. Similarly, $\boldsymbol{U}$ is also a unitary matrix, so the only transformation that affects the magnitude of $\boldsymbol{e}$ is the scaling performed by the singular value matrix $\boldsymbol{\Sigma}$. Now, we can reduce the robustness of $\boldsymbol{W}_{rec}$ to two things: the magnitude of the expansion induced by $\boldsymbol{\Sigma}$ and the effective number of directions in which $\boldsymbol{e}$ is expanded.

First, we will discuss our approach to measuring the magnitude of the expansion induced by the recurrent weights. A canonical measure of the expansion induced by a matrix is given by the spectral norm which is equivalent to the leading singular value. In this case, the spectral norm of $\boldsymbol{W}_{rec}$ tells us that $\boldsymbol{W}_{rec}(\boldsymbol{h}_t + \boldsymbol{e})$ deviates most from $\boldsymbol{W}_{rec}\boldsymbol{h}_t$ if $\boldsymbol{e}$ is the norm-1 vector parallel to the singular vector corresponding to the largest singular value of $\boldsymbol{W}_{rec}\boldsymbol{h}_t$. We can interpret this as a worst-case (i.e. adversarial) analysis of robustness. The spectral norm of $\boldsymbol{W}_{rec}$ is further tied to robustness via its relationship to the Lipschitz constant of the full network – a measure of the smoothness of the function learned by the network. In particular, a trivial upper bound for the global Lipschitz constant of a neural network is computed by multiplying the spectral norms of all the weights in

the network (Szegedy et al., 2014). However, this has been shown in many cases to be a poor proxy for network robustness due to the looseness of the bound (Huster et al., 2018). Unfortunately, since computing tight bounds for network-wide Lipschitz constants is NP-hard (Scaman & Virmaux, 2019), we maintain that it is reasonable to assume that a lower spectral norm in the recurrent weights results in a lower global Lipschitz constant *at initialization*. Of course, it is possible that during training, the spectral norm of other weights in the network increase and potentially counteract a decrease in the spectral norm of the recurrent weights induced at initialization. In practice, since the distribution shift is applied to the network input, it is also pertinent to analyze the input weights $W_{inp}$ in the recurrent module. One thing to note is the limitation of this analysis when making comparisons across architectures. In particular, recall that we have made the assumption that within a given architecture, across ranks and sparsities, models that have input and recurrent weights with lower spectral norms tend to express functions with lower Lipschitz constants (in which case a perturbation applied to the input would have less of an effect on the output). However, this assumption becomes less reasonable across architectures given the fact that the functional form varies significantly across RNNs, LSTMs and CfCs. Optimally, we would actually compute the Lipschitz constants in order to make such comparisons more viable, however we are unable to do this given the NP-hardness of the problem. Hence, while we still aim to make comparisons across architectures via an analysis of the spectral norm of the weights, it is important to acknowledge the potential limitation in this approach.

Next, we will discuss our approach to quantifying the effective number of directions in which $e$ is expanded. Note that in order to remain robust against the many potential directions the perturbation vector $e$ can lie in, it is desirable for the singular values of $W_{rec}$ to decay rapidly, as this implies that only a few singular values (i.e. only a few directions corresponding to the top singular vectors) contribute significantly to the transformation. Hence, we analyze the decay of the sorted spectrum of singular values normalized by the leading singular value as a proxy for the effective numbers of dimensions that contribute to the transformation of $e$ by $W_{rec}$. Again, it is possible that during training, the singular value spectrum of the input weights $W_{inp}$ counteracts the prior induced on the recurrent singular value spectrum at initialization and thus also must be examined. As with the spectral norm analysis discussed above, we also acknowledge the potential limitation in making comparisons of spectral decay across different recurrent architectures.

While the decay of the singular value spectrum provides a good proxy for the directionality component of robustness, it does so only for a single point in time. In actuality, we want to understand the directions the hidden state evolves in across the entire trajectory of our model in order to have a robustness measurement that takes into account all points in time. To do so, we perform a dimensionality analysis on the hidden state-space trajectories collected over the course of a simulation in the closed-loop environment. In particular, we are interested in three state-spaces: the recurrently-driven state-space, input-driven state-space and full state-space. A canonical state-space trajectory analysis collects the $h_t$ over time (i.e. the full state-space) and performs PCA in order to measure the effective dimensionality of the trajectory. We extend this analysis by decomposing the full state-space into the portion driven by the recurrent weights, $W_{rec}h_t$ and the portion driven by the input weights $W_{inp}x_t$. This allows us to disentangle the effects of the proposed parameterization of the recurrent weights into its individual effects on the recurrent and input state-spaces. Furthermore, note that this dimensionality analysis enables us to make viable comparisons across recurrent architectures whereas this is a potential limitation of analyzing only the decay of the singular value spectra in the recurrent and input weights.

So, we have now motivated the spectral norm and decay of the singular value spectrum of both the recurrent and input weights from the perspective of constructing a model that is robust to distribution shift. This was done under the framework of assuming a perturbation applied to $x_t$ which also results in a perturbation applied to $h_t$ which affects the output decision of the model at time step $t$. However, we can also ask how does this perturbation affect the model into the future: namely, how does the perturbation applied to $h_t$ affect $h_{t^*}$ for $t^* > t$. We can answer this question by understanding how information is propagated through the network across time. But note that this is precisely the intention of analyzing the time-horizon of the recurrent memory discussed in A.5.1. So, not only does modulating the recurrent memory of the model serve the purpose of enforcing a short-horizon temporal prior necessary to model the short-term causality inherent to closed-loop environment, it also makes the network more robust across the time dimension. Thus, we have two measures of robustness: one at the current point in time and another for all time points into the future.

### A.6 THEORETICAL ANALYSIS OF SPECTRAL RADIUS AND SPECTRAL NORM AT INITIALIZATION

In this section we provide proofs for the spectral radius and spectral norm, $\rho(\boldsymbol{W}), ||\boldsymbol{W}||$, respectively, for connectivity matrices $\boldsymbol{W}$ at initialization. For clarity, we iterate that $\rho(\boldsymbol{W}) = \max_i |\lambda_i|$, i.e. the largest norm of eigenvalues of $\boldsymbol{W}$, and $||\boldsymbol{W}|| = \max_i \sigma_i$, the largest singular value of $\boldsymbol{W}$. We consider sparse networks with Glorot uniform initialization and orthogonal initialization in appendices A.6.2 and A.6.4, respectively, and orthogonal low-rank matrices in appendix A.6.3. Note that in our experiments, we only consider networks with orthogonally initialized recurrent weights which we motivate further in A.10. Also, note that in the cases where we are unable to provide proof, we still perform an empirical analysis.

#### A.6.1 GLOROT UNIFORM AND SPARSE

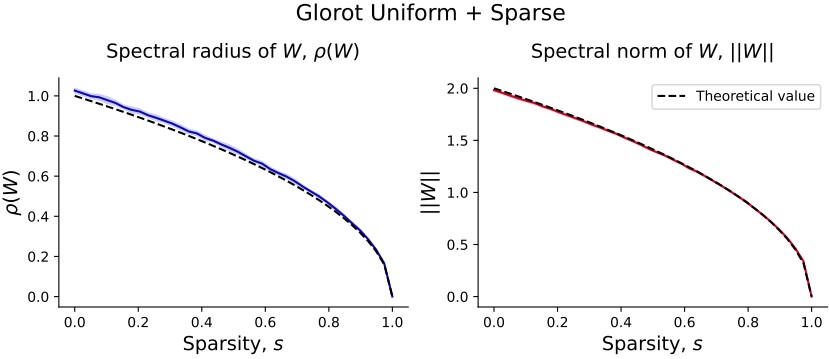

Figure 10: Spectral radius and spectral norm of uniform-sparse matrices as a function of sparsity, $s$, for matrices initialized with Glorot uniform initialization, with $n = 512$. We additionally plot the theoretical predicted value for large $n$.

We consider weight matrices $\boldsymbol{W}$ with dimension $n \times n$ and consider the limit as $n \to \infty$. Recall that Glorot uniform initialization scheme initializes weights with entries iid $\boldsymbol{W}_{ij} \sim$ Unif $\left(-\frac{\sqrt{3}}{\sqrt{n}}, +\frac{\sqrt{3}}{\sqrt{n}}\right)$, resulting in an entry-wise variance of $\frac{1}{n}$. We generate a sparse matrix by element-wise multiplying a matrix, $\boldsymbol{W}^0$ sampled from the Glorot uniform initialization by a sparsity map $\boldsymbol{M}$, where $\boldsymbol{M}_{ij} \sim$ Bernoulli$(p = 1 - s)$, with $s$ being the sparse factor. Our final weight matrix is given by $\boldsymbol{W} = \boldsymbol{W}^0 \odot \boldsymbol{M}$. For sparsity $s$, the resulting variance of entries is given by $\frac{1-s}{n}$. As the entries of $\boldsymbol{W}$ are all iid, we can apply the Girko-Ginibri circular law for large $n$, which states that the eigenvalues of $\boldsymbol{W}$ converge to a uniform disk in the complex plane with radius $\sqrt{1-s}$, so we expect $\rho(s) = \sqrt{1-s}$. This analysis follows closely with that of Herbert & Ostojic (2022).

For $||\boldsymbol{W}||$, we apply the Marchenko-Pastur law, which states that the distribution of eigenvalues of $\frac{1}{n}\boldsymbol{A}\boldsymbol{A}^T$ values converges to $p_{\boldsymbol{A}\boldsymbol{A}^T}(\lambda) = \frac{1}{2\pi\sigma^2}\frac{\sqrt{(\lambda_+ - \lambda)(\lambda - \lambda_-)}}{\lambda}\mathbf{1}_{\lambda \in [\lambda_-, \lambda_+]}$, with $\lambda_\pm = \sigma^2(1 \pm 1)^2$, if $\boldsymbol{A}$ has entries iid from a distribution with zero mean and variance $\sigma^2$. For our setting we let $\boldsymbol{A} = \sqrt{n}\boldsymbol{W}$, so $\sigma^2 = 1 - s$. We see that the maximal eigenvalue of $\boldsymbol{A}\boldsymbol{A}^T$ is thus $\lambda_+ = 4(1-s)$, and so the upper bound for the largest singular value of $\boldsymbol{W}$ is $||\boldsymbol{W}|| = 2\sqrt{(1-s)}$. We observe good empirical agreement with these values in Figure 10.

#### A.6.2 ORTHOGONAL AND SPARSE

In the orthogonal-sparse case, we cannot apply the same arguments in appendix A.6.4, as the entries of an orthogonal matrix are no longer iid. However, we note that the entries of orthogonal matrices are approximately Gaussian when considered individually (Życzkowski & Sommers, 1999). Thus, we expect with high sparsity, the correlations between entries break down and the entries of the matrix behave iid. Thus, we expect for large values of $s$, $\rho(\boldsymbol{W})$ and $||\boldsymbol{W}||$ to have the same behavior

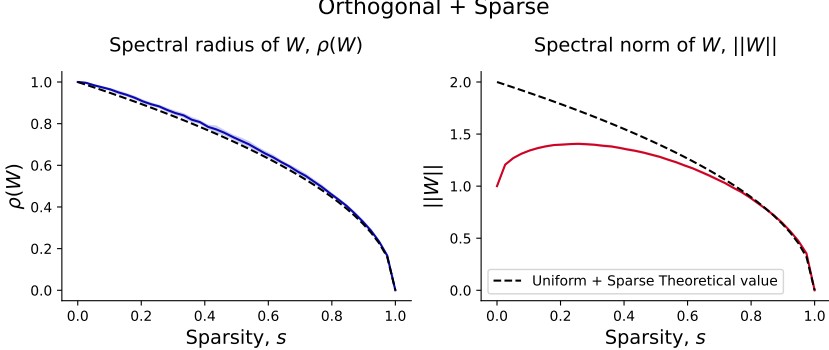

Figure 11: Spectral radius and spectral norm of orthogonal-sparse matrices as a function of sparsity, $s$, with $n = 512$. We additionally plot the theoretical predicted value for large $n$ for the uniform-sparse initialization. We see that orthogonal-sparse matrices behave like uniform-sparse ones at high sparsities.

as in appendix A.6.4. We verify this empirically in Figure 11, where we see that for large values of $s$, $\rho(\boldsymbol{W}) \approx \sqrt{1-s}$ and $||\boldsymbol{W}|| \approx 2\sqrt{1-s}$, as with appendix A.6.4. Interestingly, $||\boldsymbol{W}||$ is non-monotonic in $s$. Calculating the exact forms of $\rho(\boldsymbol{W})$ and $||\boldsymbol{W}||$ is an interesting direction for future work.

### A.6.3 ORTHOGONAL AND LOW RANK

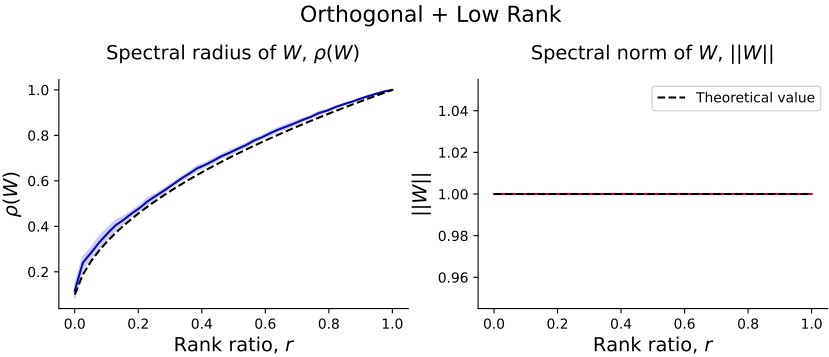

Figure 12: Spectral radius and spectral norm of orthogonal-low-rank matrices as a function of rank ratio, $r$, with $n = 512$. We additionally plot the theoretical predicted value for large $n$.

Recall that to generate our low rank connectivity matrix $\boldsymbol{W}$, we first generate an orthogonal matrix $\boldsymbol{W}^0$, then perform an SVD of it $\boldsymbol{W} = \boldsymbol{U}\boldsymbol{S}\boldsymbol{V}^T = \sum_i^n \sum_i^n \boldsymbol{u}_i \boldsymbol{v}_i^T$, and then truncate it at rank k to get $\boldsymbol{W} = \sum_i^k \boldsymbol{u}_i \boldsymbol{v}_i^T$ with $k \leq n$. For $||\boldsymbol{W}||$, we note that because all the singular values are 1, truncating the SVD does not change the maximum singular value so $||\boldsymbol{W}|| = 1$ for all rank ratios $r = \frac{k}{n}$.

For $\rho(\boldsymbol{W})$, we note that one valid SVD of $\boldsymbol{W}^0$ is $\boldsymbol{W}^0 = \boldsymbol{W}^0\boldsymbol{I}\boldsymbol{I}$, with $\boldsymbol{U} = \boldsymbol{W}^0$, $\boldsymbol{S} = \boldsymbol{I}$ and $\boldsymbol{V} = \boldsymbol{I}$. It is sufficient to consider this particular SVD, as other SVDs correspond to arbitrary rotations of $\boldsymbol{U}$ and $\boldsymbol{V}$, which will not affect the final eigenvalue distribution in expectation. For this particular SVD, $\boldsymbol{W} = [\boldsymbol{W}^0_1, \boldsymbol{W}^0_2, \dots, \boldsymbol{W}^0_k, 0, \dots, 0]$, i.e. we take the first $k$ columns of $\boldsymbol{W}^0$, and replace the remaining entries with 0. Due to the large zero block in this matrix, $\boldsymbol{W}$ has the same non-zero eigenvectors and eigenvalues as the $k \times k$ principal submatrix of $\boldsymbol{W}^0$: $\boldsymbol{W}^0_{1:k,1:k}$[1]. Życzkowski & Sommers (1999) studies the eigenvalues of the principal submatrix of random orthogonal matrices

---

[1]for eigenvectors we need to concatenate the remaining $n - k$ zeros to have the correct dimension

and shows that the distribution of these eigenvalues has mean $\mathbb{E}[\lambda] = \sqrt{r}$ for rank ratio $r = \frac{k}{n}$. Furthermore, $\mathbf{Var}[\lambda] \to 0$ as $n \to \infty$ at fixed $r$ (Życzkowski & Sommers, 1999), so we have $\rho(\boldsymbol{W}) = \sqrt{r}$ for large $n$. Empirical verification of this is provided in Figure 12 for $n = 512$.

### A.6.4 GLOROT UNIFORM AND LOW RANK

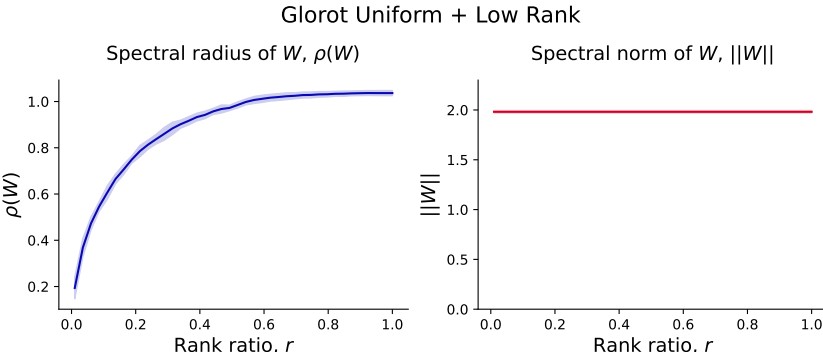

Figure 13: Spectral radius and spectral norm of uniform-low-rank matrices as a function of rank ratio, $r$, with $n = 512$.

As with the orthogonal-low-rank case, low-rank approximation does not affect the spectral norm of $\boldsymbol{W}$ when using a Glorot Uniform initialization scheme. Proving results for the spectral radius of low-rank uniform initialization is more difficult due to the more complex eigenstructure of uniformly initialized matrices compared to orthogonal ones. Thus, we leave it to future work to formally prove the relationship between rank ratio and spectral radius. We observe empirically in Figure 13 that spectral radius is increasing with rank ratio, like with the orthogonal-low-rank case.

### A.7 SINGULAR VALUE SPECTRUM AND EIGENSPECTRUM AT INITIALIZATION

Recall in A.6, we provided theoretical results for the spectral norm and spectral radius of randomly initialized matrices as a function of rank and sparsity in order to provide theoretical guarantees regarding the proposed parameterization of the recurrent weights $\boldsymbol{W}_{rec}(r, s)$ at initialization. As explained in A.5.2, we also motivated the parameterization with respect to the entire the spectrum of singular values due to its connection with robustness. Our theoretical results regarding the spectral norm as a function of rank and sparsity do not extend to the full spectrum of singular values. So, here we provide empirical evidence regarding the decay of the singular value spectrum of the recurrent weights as a function of rank and sparsity (we also provide plots of the decay of the eigenspectrum of the recurrent weights to demonstrate the similarity between the two related, yet distinct spectra).

### A.7.1 ORTHOGONAL INITIALIZATION

Here, we examine the singular value spectrum for the orthogonally initialized recurrent weights across various ranks and sparsities. We note that as sparsity increases, the rate at which the singular value spectrum decays decrease (Figure 14). As discussed in A.5.2, this raises the effective dimensionality of the transformation induced by the recurrent weights on the hidden-state vector. Similarly, as rank increases, the rate at which the singular value spectrum decays also decreases (Figure 14). This is interesting as increasing sparsity removes parameters from the model, whereas increases rank adds parameters to the model. Hence, we observe that pruning by increasing sparsity and pruning by decreasing rank are distinct with respect to the decay each induces in the singular value spectrum of the recurrent weights. Furthermore, note that we observe the exact same patterns with respect to the recurrent eigenspectra as well (Figure 15).

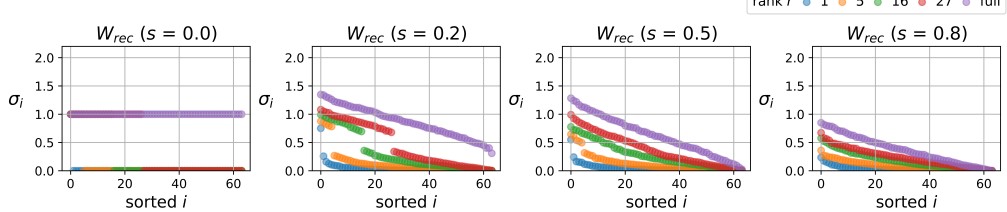

Figure 14: Singular value spectra of orthogonally initialized recurrent weights across various ranks and sparsities sorted in decreasing order.

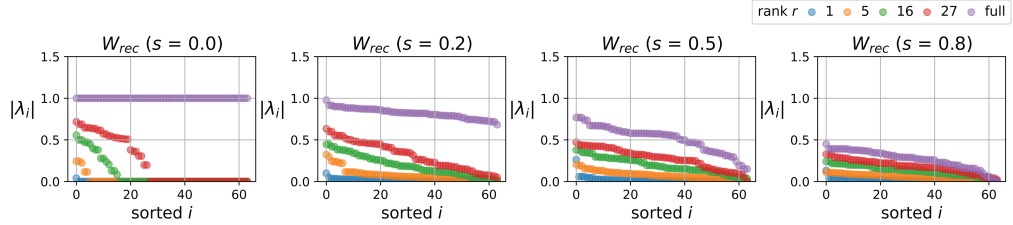

Figure 15: Eigenspectra of orthogonally initialized recurrent weights across various ranks and sparsities sorted in decreasing order.

### A.7.2 GLOROT UNIFORM INITIALIZATION

Here, we examine the singular value spectrum for the Glorot uniform initialized recurrent weights across various ranks and sparsities. The decay patterns observed with respect to the singular value and eigenvalue spectra in the orthogonally initialized weights hold here as well (Figures 16, 17).

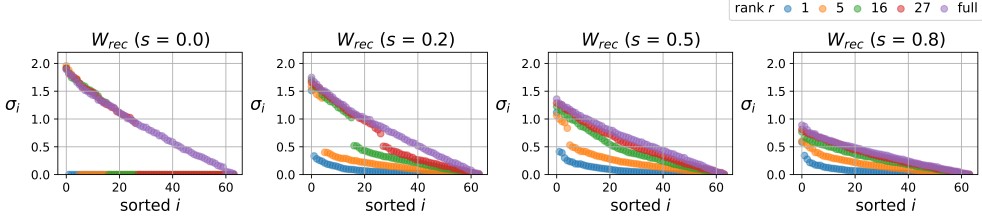

Figure 16: Singular value spectra of Glorot uniform initialized recurrent weights across various ranks and sparsities sorted in decreasing order.

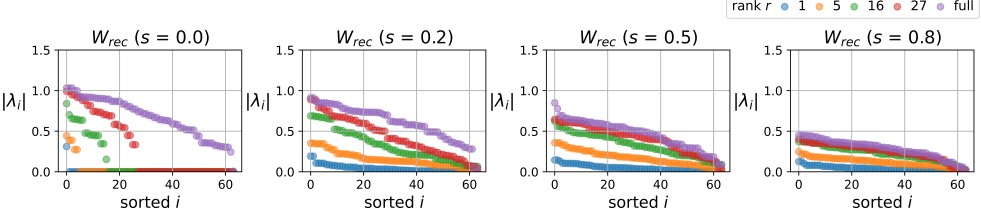

Figure 17: Eigenspectra of Glorot uniform initialized recurrent weights across various ranks and sparsities sorted in decreasing order.

## A.8 TRAINED NETWORK DYNAMICS RESULTS

In this section, we provide plots for various analyses we conducted on the trained networks that were not included in the main portion of the paper.

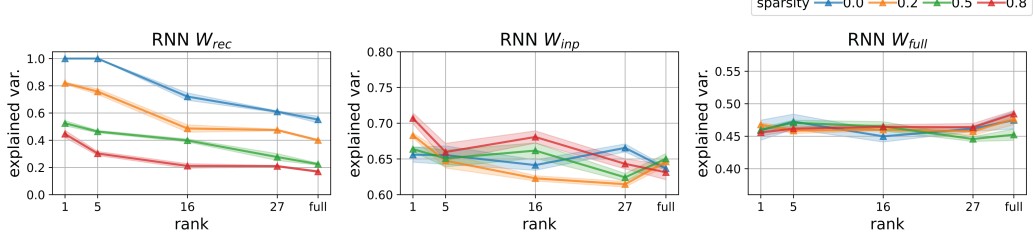

Figure 18: Effective dimensionality of RNN state-space trajectories across various ranks and sparsities. Note that the dynamics of RNNs are higher-dimensional than the dynamics we observe in CfCs. This, alongside the disparity in spectral norm, offers intuition as to why we find that CfCs tend to outperform RNNs under distribution shift.

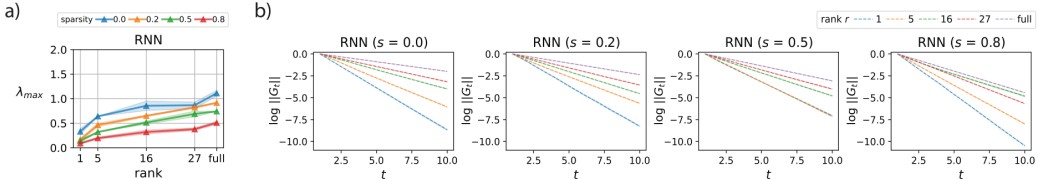

Figure 19: a) Spectral radius of recurrent weights $W_{rec}(r, s)$ in trained RNN networks across ranks and sparsities. b) Frobenius norm of recurrent gradients $G_t$ as a function of time. Note that the norms are plotted in log space and translated to decay from $0$ to enable easier comparison. For details on the computation, refer to A.11.

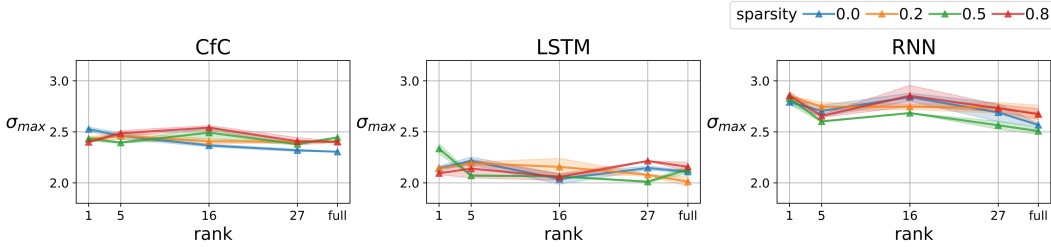

Figure 20: Spectral norm of the input weights across models, ranks and sparsities.

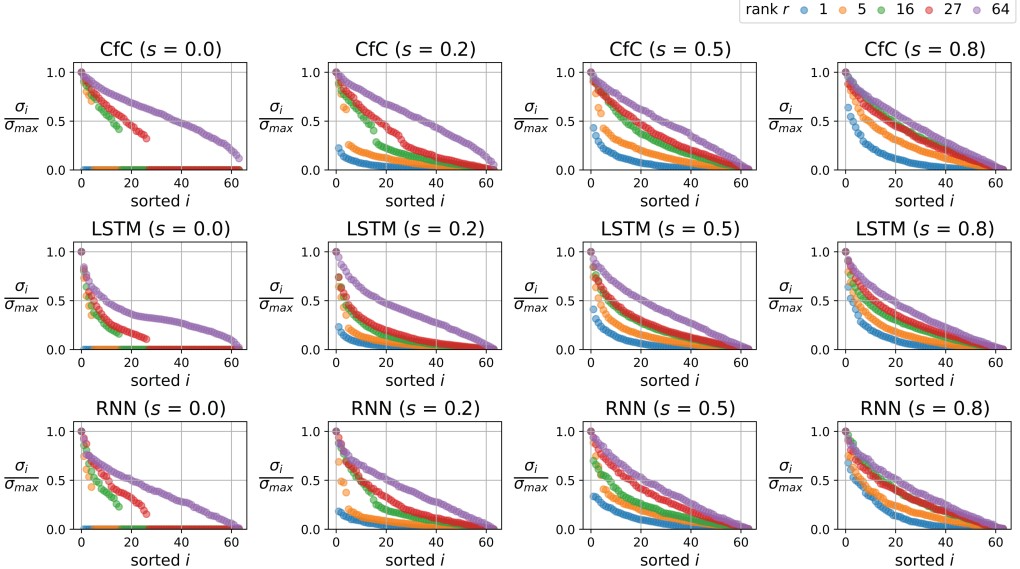

Figure 21: Decay of the recurrent singular value spectrum across models, ranks and sparsities as measured by normalizing the sorted spectrum by the spectral norm. We find that the prior induced at initialization holds at convergence: namely, as sparsity increases the rate of decay of the spectrum decreases and as rank decreases the rate of decay of the spectrum increases.

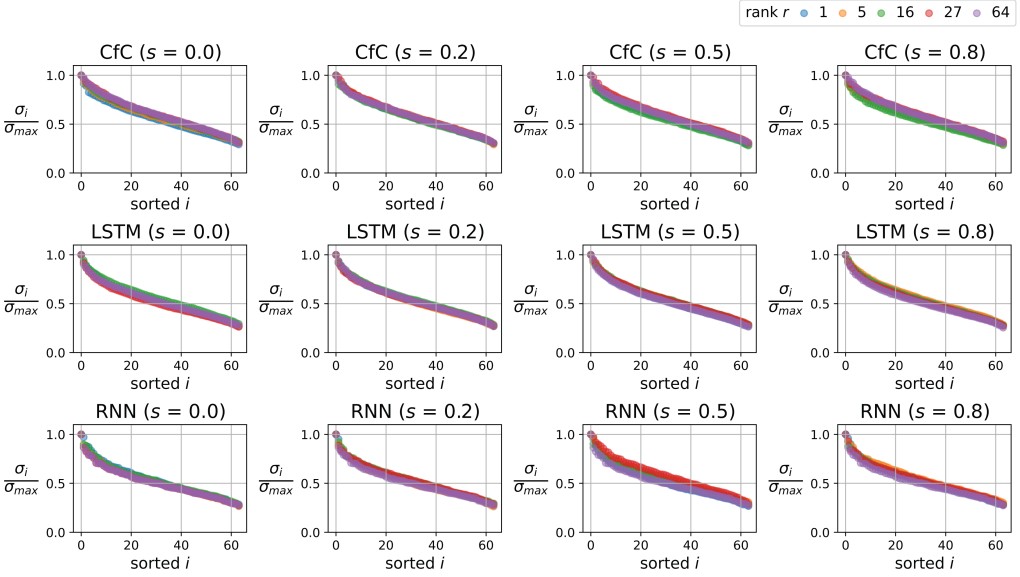

Figure 22: Decay of the input singular value spectrum across models, ranks and sparsities as measured by normalizing the sorted spectrum by the spectral norm. We find that the decay is quite similar across models, sparsities and ranks, which aligns with many of our findings suggesting that the input weights are little affected by changes in the rank and sparsity of the recurrent weights.

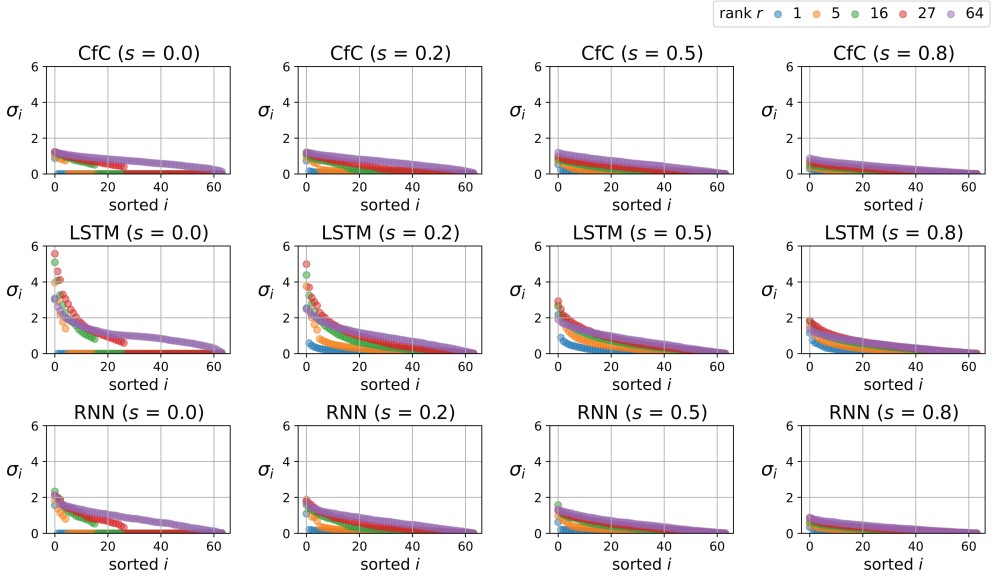

Figure 23: Singular value spectrum of the recurrent weights of the trained models across various ranks and sparsities sorted in decreasing order.

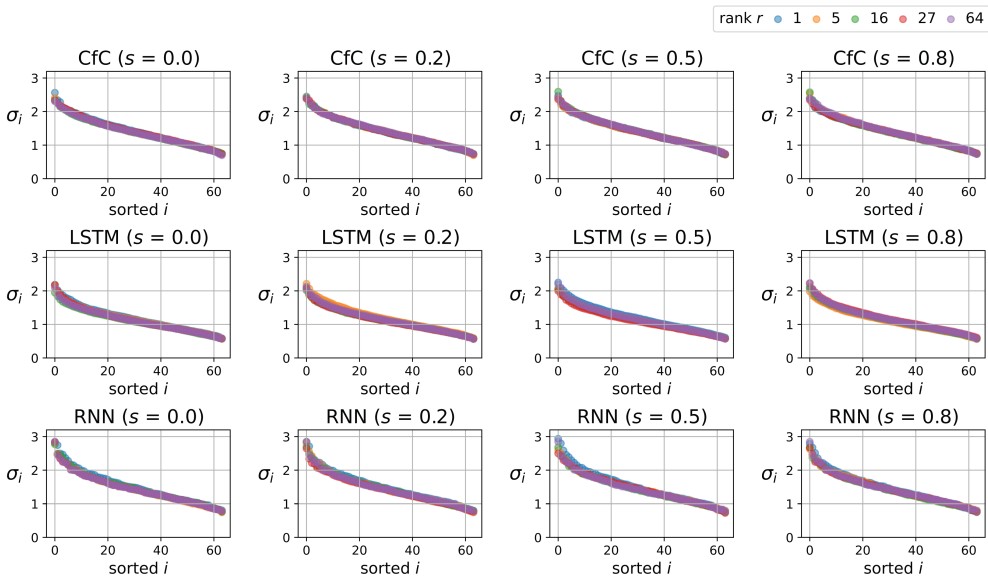

Figure 24: Singular value spectrum of the input weights of the trained models across various ranks and sparsities sorted in decreasing order.

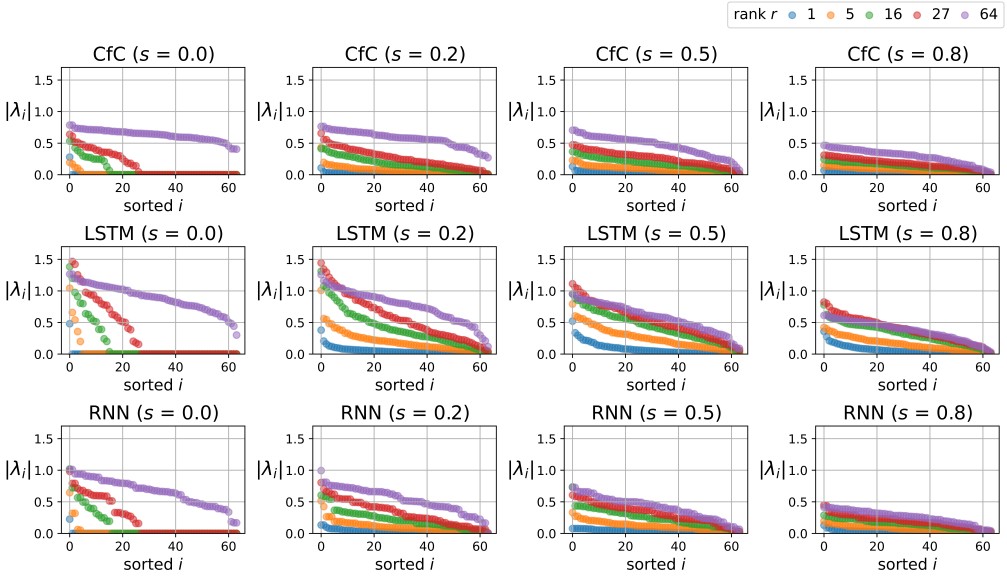

Figure 25: Eigenspectrum of the recurrent weights of the trained models across various ranks and sparsities sorted in decreasing order of magnitude.

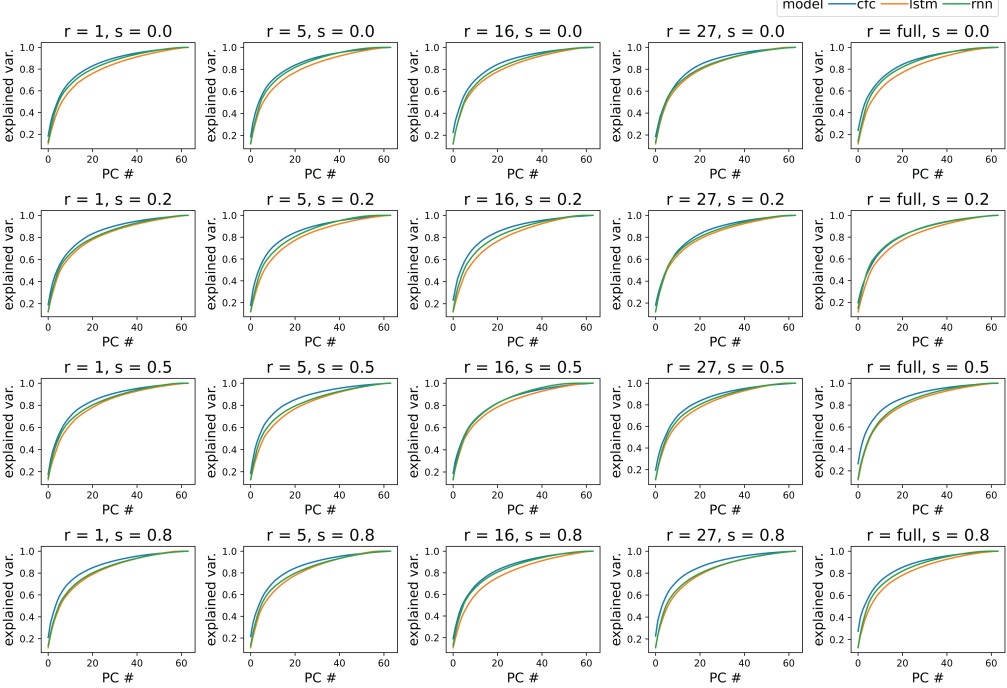

Figure 26: Dimensionality analysis of full state-space trajectories collected from the online, closed-loop in-distribution simulation analyzed in Figure 6. Here, we plot the explained variance curve over all principal components to provide a more complete view of the effective dimensionality of the trajectories, as opposed to only the top 5 PCs.

## A.9 FULL RESULTS

### A.9.1 ONLINE AND OFFLINE PERFORMANCE

Here, we provide offline validation/test loss, online in-distribution rewards and online rewards under distribution shift for each Gym environment (Brockman et al., 2016) that we ran experiments on: Seaquest, Alien and HalfCheetah. For each environment, we considered 5 types of models: CfCs, RNNs, LSTMs, GRUs and CNNs. For each of the recurrent architectures we considered models of various 5 different ranks and 4 different sparsities. Full details on the experimental setup are given in A.11.

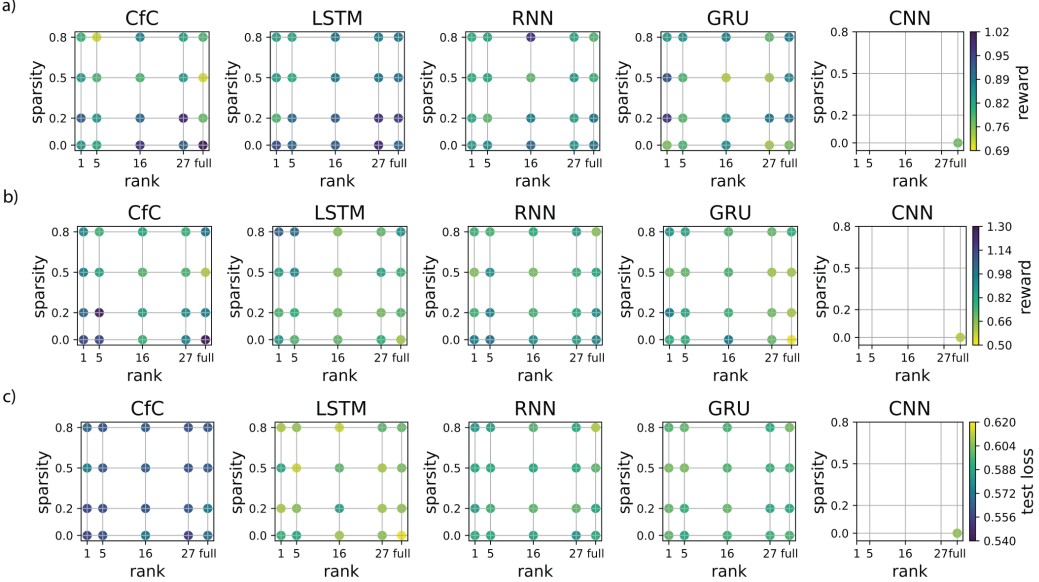

Figure 27: Online and offline performance of recurrent networks under different ranks and sparsities in the Seaquest environment. Note that online performance of Seaquest was presented in the main portion of the paper, but here we include it alongside the offline losses evaluated on validation/test set. a) In-distribution rewards in the online, closed-loop setting normalized by the rewards obtained by the expert in-distribution. b) Rewards averaged across 5 distribution shifts, normalized by the rewards obtained by the expert under distribution shift. c) Loss of model evaluated on a validation/test set.

| Model | Rank | Sparsity | | | |
|-------|------|----------|---|---|---|
| | | 0 | 0.2 | 0.5 | 0.8 |
| CfC | 1 | 0.86 (0.03) | 0.93 (0.04) | 0.88 (0.03) | 0.8 (0.05) |
| | 5 | 0.81 (0.06) | 0.86 (0.05) | 0.79 (0.06) | 0.71 (0.06) |
| | 16 | 1.01 (0.06) | 0.92 (0.06) | 0.79 (0.05) | 0.9 (0.05) |
| | 27 | 0.91 (0.06) | 0.97 (0.04) | 0.88 (0.06) | 0.83 (0.06) |
| | full | 0.97 (0.04) | 0.76 (0.08) | 0.7 (0.05) | 0.76 (0.06) |
| LSTM | 1 | 0.99 (0.04) | 0.83 (0.04) | 0.9 (0.06) | 0.92 (0.05) |
| | 5 | 0.94 (0.03) | 0.94 (0.05) | 0.88 (0.06) | 0.88 (0.05) |
| | 16 | 0.94 (0.04) | 0.94 (0.07) | 0.9 (0.06) | 0.9 (0.07) |
| | 27 | 1.0 (0.03) | 0.99 (0.05) | 0.9 (0.06) | 0.88 (0.06) |
| | full | 0.9 (0.05) | 0.98 (0.03) | 0.9 (0.06) | 0.9 (0.06) |
| RNN | 1 | 0.94 (0.06) | 0.92 (0.06) | 0.89 (0.05) | 0.9 (0.05) |
| | 5 | 0.96 (0.05) | 0.86 (0.05) | 0.94 (0.06) | 0.92 (0.06) |
| | 16 | 0.94 (0.06) | 0.9 (0.05) | 0.88 (0.06) | 1.07 (0.02) |
| | 27 | 0.88 (0.05) | 0.83 (0.05) | 1.01 (0.06) | 0.91 (0.03) |
| | full | 0.92 (0.04) | 0.92 (0.04) | 0.92 (0.05) | 0.86 (0.06) |
| GRU | 1 | 0.81 (0.06) | 1.04 (0.04) | 1.0 (0.05) | 0.9 (0.04) |
| | 5 | 0.85 (0.04) | 0.84 (0.06) | 0.85 (0.06) | 0.91 (0.04) |
| | 16 | 0.92 (0.06) | 0.92 (0.05) | 0.78 (0.05) | 0.94 (0.05) |
| | 27 | 0.79 (0.05) | 0.95 (0.06) | 0.79 (0.06) | 0.84 (0.05) |
| | full | 0.82 (0.04) | 0.95 (0.06) | 0.93 (0.06) | 0.94 (0.05) |
| CNN | 1 | — | — | — | — |
| | 5 | — | — | — | — |
| | 16 | — | — | — | — |
| | 27 | — | — | — | — |
| | full | 0.77 (0.05) | — | — | — |

Table 1: Mean episodic in-distribution rewards in Seaquest environment normalized by rewards obtained by expert policy. Normalized rewards are given $\pm 1$ SE which is shown in parentheses.

| Model | Rank | Sparsity | | | |
|---|---|---|---|---|---|
| | | 0 | 0.2 | 0.5 | 0.8 |
| CfC | 1 | 1.12 (0.05) | 1.03 (0.03) | 0.95 (0.02) | 0.93 (0.03) |
| | 5 | 1.12 (0.03) | 1.26 (0.03) | 0.83 (0.02) | 0.76 (0.03) |
| | 16 | 0.76 (0.03) | 0.85 (0.03) | 0.73 (0.03) | 0.8 (0.03) |
| | 27 | 0.95 (0.02) | 0.91 (0.03) | 0.74 (0.02) | 0.77 (0.03) |
| | full | 1.28 (0.03) | 0.95 (0.03) | 0.59 (0.04) | 0.98 (0.03) |
| LSTM | 1 | 0.77 (0.03) | 0.68 (0.04) | 0.76 (0.03) | 0.91 (0.02) |
| | 5 | 0.67 (0.04) | 0.71 (0.04) | 0.82 (0.03) | 0.89 (0.02) |
| | 16 | 0.66 (0.05) | 0.64 (0.04) | 0.62 (0.05) | 0.62 (0.05) |
| | 27 | 0.7 (0.04) | 0.63 (0.05) | 0.74 (0.04) | 0.66 (0.05) |
| | full | 0.58 (0.05) | 0.67 (0.04) | 0.69 (0.04) | 0.78 (0.05) |
| RNN | 1 | 0.78 (0.02) | 0.96 (0.02) | 0.7 (0.04) | 0.87 (0.02) |
| | 5 | 0.78 (0.02) | 0.82 (0.02) | 0.7 (0.03) | 0.83 (0.03) |
| | 16 | 0.96 (0.02) | 0.76 (0.03) | 0.73 (0.03) | 0.73 (0.03) |
| | 27 | 0.71 (0.04) | 0.68 (0.04) | 0.62 (0.04) | 0.71 (0.04) |
| | full | 0.51 (0.03) | 0.61 (0.04) | 0.61 (0.04) | 0.78 (0.04) |
| GRU | 1 | 0.8 (0.05) | 0.67 (0.05) | 0.61 (0.04) | 0.67 (0.04) |
| | 5 | 0.84 (0.05) | 0.8 (0.05) | 0.76 (0.04) | 0.66 (0.04) |
| | 16 | 0.76 (0.04) | 0.66 (0.04) | 0.62 (0.04) | 0.69 (0.04) |
| | 27 | 0.73 (0.05) | 0.69 (0.04) | 0.67 (0.04) | 0.73 (0.04) |
| | full | 0.78 (0.04) | 0.76 (0.04) | 0.7 (0.04) | 0.6 (0.04) |
| CNN | 1 | — | — | — | — |
| | 5 | — | — | — | — |
| | 16 | — | — | — | — |
| | 27 | — | — | — | — |
| | full | 0.58 (0.04) | — | — | — |

Table 2: Mean episodic rewards under distribution shift in Seaquest environment normalized by performance of the expert policy under distribution shift. Rewards are given $\pm 1$ SE which is shown in parentheses.

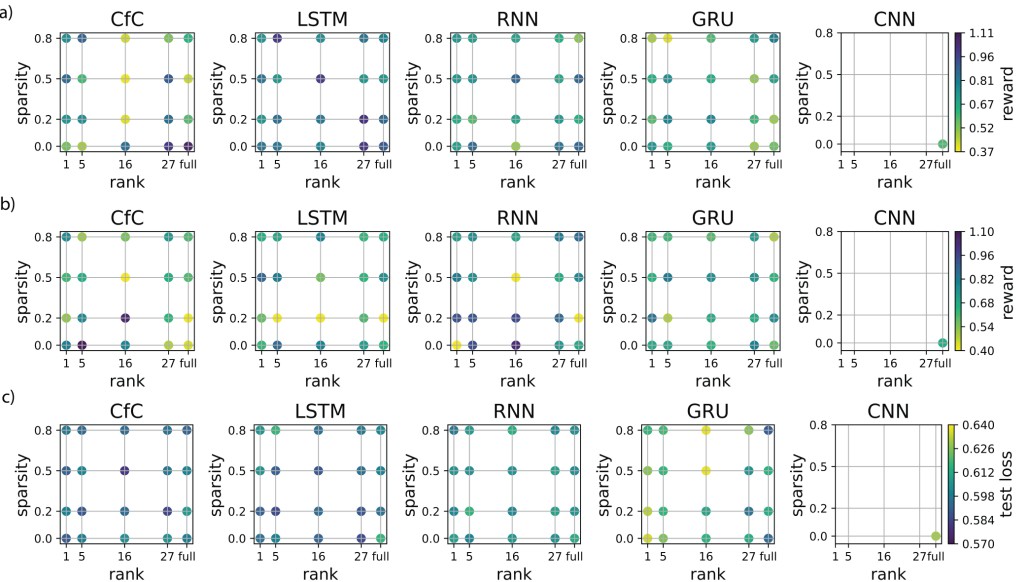

Figure 28: Online and offline performance of recurrent networks under different ranks and sparsities in the Alien environment. a) In-distribution rewards in the online, closed-loop setting normalized by rewards obtained by the expert in-distribution. a) In-distribution rewards in the online, closed-loop setting normalized by the rewards obtained by the expert in-distribution. b) Rewards averaged across 5 distribution shifts, normalized by the rewards obtained by the expert under distribution shift. c) Loss of model evaluated on a validation/test set.

| Model | Rank | Sparsity | | | |
|---|---|---|---|---|---|
| | | 0 | 0.2 | 0.5 | 0.8 |
| CfC | 1 | 0.59 (0.07) | 0.81 (0.14) | 0.95 (0.11) | 0.83 (0.19) |
| | 5 | 0.52 (0.1) | 0.84 (0.14) | 0.66 (0.13) | 0.98 (0.14) |
| | 16 | 0.94 (0.14) | 0.45 (0.05) | 0.42 (0.05) | 0.47 (0.04) |
| | 27 | 1.1 (0.18) | 0.94 (0.19) | 1.01 (0.17) | 0.58 (0.08) |
| | full | 1.2 (0.21) | 0.64 (0.12) | 0.47 (0.07) | 0.7 (0.11) |
| LSTM | 1 | 0.92 (0.12) | 0.8 (0.12) | 0.92 (0.12) | 0.81 (0.12) |
| | 5 | 0.9 (0.12) | 0.93 (0.15) | 0.8 (0.15) | 1.1 (0.18) |
| | 16 | 0.87 (0.13) | 0.9 (0.12) | 1.09 (0.13) | 0.84 (0.16) |
| | 27 | 1.11 (0.14) | 1.12 (0.15) | 0.8 (0.14) | 0.92 (0.12) |
| | full | 1.0 (0.14) | 0.95 (0.14) | 1.15 (0.12) | 0.88 (0.17) |
| RNN | 1 | 0.77 (0.12) | 0.73 (0.11) | 0.8 (0.1) | 0.8 (0.11) |
| | 5 | 0.96 (0.11) | 0.64 (0.11) | 0.82 (0.1) | 0.78 (0.14) |
| | 16 | 0.54 (0.1) | 0.76 (0.11) | 0.97 (0.06) | 0.76 (0.11) |
| | 27 | 0.92 (0.11) | 0.79 (0.12) | 0.75 (0.1) | 0.71 (0.11) |
| | full | 0.97 (0.09) | 0.71 (0.13) | 0.97 (0.11) | 0.57 (0.1) |
| GRU | 1 | 0.81 (0.11) | 0.64 (0.11) | 0.71 (0.11) | 0.52 (0.11) |
| | 5 | 0.72 (0.09) | 0.87 (0.12) | 0.83 (0.12) | 0.44 (0.06) |
| | 16 | 0.8 (0.1) | 0.86 (0.11) | 0.72 (0.1) | 0.64 (0.12) |
| | 27 | 0.54 (0.1) | 0.67 (0.11) | 0.55 (0.1) | 0.79 (0.11) |
| | full | 0.6 (0.1) | 0.54 (0.12) | 0.77 (0.14) | 0.89 (0.11) |
| CNN | 1 | — | — | — | — |
| | 5 | — | — | — | — |
| | 16 | — | — | — | — |
| | 27 | — | — | — | — |
| | full | 0.64 (0.09) | — | — | — |

Table 3: Mean episodic in-distribution rewards in Alien environment normalized by rewards obtained by expert policy. Normalized rewards are given $\pm 1$ SE which is shown in parentheses.

| Model | Rank | Sparsity | | | |
|---|---|---|---|---|---|
| | | 0 | 0.2 | 0.5 | 0.8 |
| CfC | 1 | 0.81 (0.04) | 0.52 (0.02) | 0.62 (0.03) | 0.77 (0.05) |
| | 5 | 1.1 (0.02) | 0.78 (0.02) | 0.67 (0.02) | 0.51 (0.03) |
| | 16 | 0.8 (0.02) | 1.05 (0.05) | 0.3 (0.02) | 0.55 (0.03) |
| | 27 | 0.48 (0.02) | 0.64 (0.02) | 0.66 (0.02) | 0.73 (0.01) |
| | full | 0.44 (0.01) | 0.41 (0.02) | 0.6 (0.03) | 0.6 (0.02) |
| LSTM | 1 | 0.63 (0.03) | 0.58 (0.05) | 0.88 (0.05) | 0.66 (0.06) |
| | 5 | 0.87 (0.06) | 0.08 (0.0) | 0.81 (0.06) | 0.69 (0.05) |
| | 16 | 0.81 (0.06) | 0.08 (0.0) | 0.56 (0.03) | 0.8 (0.04) |
| | 27 | 0.72 (0.06) | 0.6 (0.05) | 0.64 (0.05) | 0.63 (0.05) |
| | full | 0.65 (0.06) | 0.3 (0.01) | 0.76 (0.06) | 0.66 (0.04) |
| RNN | 1 | 0.68 (0.03) | 0.84 (0.05) | 0.65 (0.03) | 0.62 (0.03) |
| | 5 | 0.69 (0.06) | 0.48 (0.04) | 0.82 (0.05) | 0.62 (0.04) |
| | 16 | 0.65 (0.03) | 0.68 (0.05) | 0.76 (0.05) | 0.6 (0.04) |
| | 27 | 0.8 (0.04) | 0.67 (0.05) | 0.74 (0.06) | 0.72 (0.05) |
| | full | 0.58 (0.04) | 0.77 (0.04) | 0.64 (0.05) | 0.49 (0.02) |
| GRU | 1 | 0.1 (0.0) | 0.94 (0.05) | 0.82 (0.03) | 0.71 (0.04) |
| | 5 | 0.95 (0.05) | 0.92 (0.08) | 0.76 (0.05) | 0.77 (0.05) |
| | 16 | 1.04 (0.03) | 0.95 (0.05) | 0.08 (0.0) | 0.69 (0.03) |
| | 27 | 0.73 (0.04) | 0.87 (0.05) | 0.67 (0.05) | 0.82 (0.07) |
| | full | 0.69 (0.05) | 0.08 (0.0) | 0.78 (0.03) | 0.86 (0.05) |
| CNN | 1 | — | — | — | — |
| | 5 | — | — | — | — |
| | 16 | — | — | — | — |
| | 27 | — | — | — | — |
| | full | 0.67 (0.04) | — | — | — |

Table 4: Mean episodic rewards under distribution shift in Alien environment normalized by performance of the expert policy under distribution shift. Rewards are given $\pm 1$ SE which is shown in parentheses.

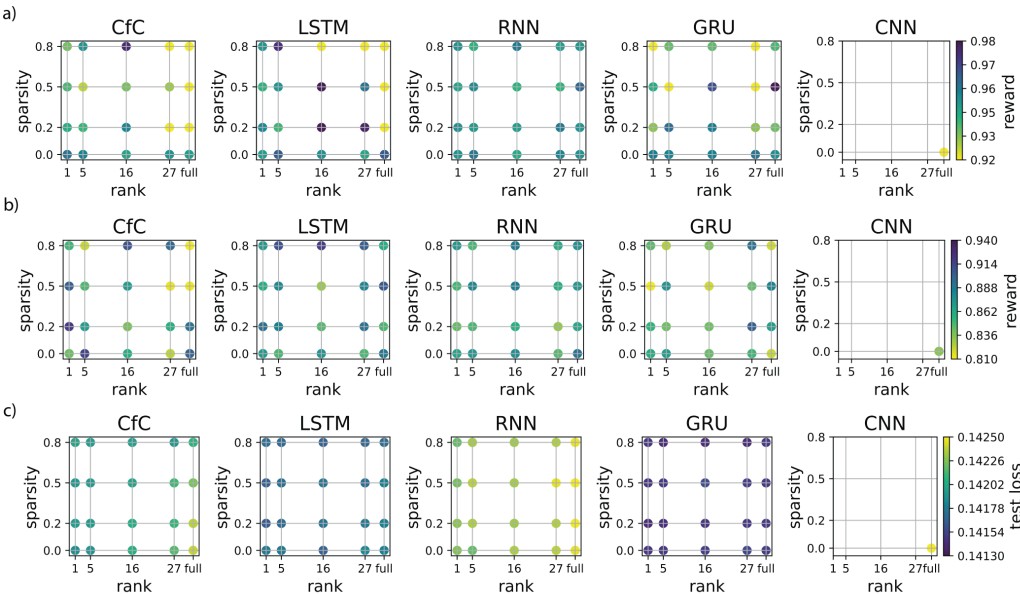

Figure 29: Online and offline performance of recurrent networks under different ranks and sparsities in the HalfCheetah environment. a) In-distribution rewards in the online, closed-loop setting normalized by the rewards obtained by the expert in-distribution. b) Rewards averaged across 5 distribution shifts, normalized by the rewards obtained by the expert under distribution shift. c) Loss of model evaluated on a validation/test set.

| Model | Rank | Sparsity | | | |
|---|---|---|---|---|---|
| | | 0 | 0.2 | 0.5 | 0.8 |
| CfC | 1 | 0.945 (0.005) | 0.929 (0.003) | 0.924 (0.004) | 0.92 (0.003) |
| | 5 | 0.941 (0.003) | 0.926 (0.003) | 0.91 (0.002) | 0.936 (0.002) |
| | 16 | 0.932 (0.002) | 0.941 (0.003) | 0.919 (0.002) | 0.963 (0.005) |
| | 27 | 0.937 (0.004) | 0.874 (0.004) | 0.914 (0.003) | 0.883 (0.004) |
| | full | 0.934 (0.003) | 0.816 (0.006) | 0.877 (0.002) | 0.847 (0.003) |
| LSTM | 1 | 0.933 (0.003) | 0.94 (0.003) | 0.927 (0.003) | 0.936 (0.003) |
| | 5 | 0.95 (0.003) | 0.925 (0.004) | 0.937 (0.002) | 0.962 (0.003) |
| | 16 | 0.937 (0.003) | 0.991 (0.003) | 0.968 (0.005) | 0.905 (0.003) |
| | 27 | 0.931 (0.003) | 0.965 (0.004) | 0.945 (0.003) | 0.872 (0.003) |
| | full | 0.951 (0.003) | 0.875 (0.003) | 0.889 (0.008) | 0.842 (0.009) |
| RNN | 1 | 0.943 (0.004) | 0.917 (0.007) | 0.93 (0.005) | 0.903 (0.003) |
| | 5 | 0.941 (0.003) | 0.946 (0.003) | 0.885 (0.003) | 0.921 (0.003) |
| | 16 | 0.941 (0.002) | 0.942 (0.003) | 0.954 (0.003) | 0.922 (0.004) |
| | 27 | 0.944 (0.003) | 0.917 (0.003) | 0.9 (0.029) | 0.861 (0.006) |
| | full | 0.938 (0.003) | 0.92 (0.002) | 0.971 (0.003) | 0.924 (0.021) |
| GRU | 1 | 0.937 (0.002) | 0.938 (0.004) | 0.932 (0.003) | 0.937 (0.002) |
| | 5 | 0.944 (0.002) | 0.935 (0.003) | 0.938 (0.003) | 0.927 (0.003) |
| | 16 | 0.93 (0.003) | 0.934 (0.003) | 0.927 (0.003) | 0.943 (0.003) |
| | 27 | 0.94 (0.002) | 0.943 (0.003) | 0.926 (0.011) | 0.934 (0.001) |
| | full | 0.937 (0.004) | 0.936 (0.003) | 0.949 (0.003) | 0.935 (0.002) |
| CNN | 1 | — | — | — | — |
| | 5 | — | — | — | — |
| | 16 | — | — | — | — |
| | 27 | — | — | — | — |
| | full | 0.90 (0.003) | — | — | — |

Table 5: Mean episodic in-distribution rewards in HalfCheetah environment normalized by rewards obtained by expert policy. Normalized rewards are given $\pm 1$ SE which is shown in parentheses.

| Model | Rank | Sparsity | | | |
|---|---|---|---|---|---|
| | | 0 | 0.2 | 0.5 | 0.8 |
| CfC | 1 | 0.842 (0.009) | 0.919 (0.001) | 0.903 (0.003) | 0.852 (0.004) |
| | 5 | 0.915 (0.002) | 0.869 (0.007) | 0.848 (0.004) | 0.82 (0.005) |
| | 16 | 0.86 (0.007) | 0.838 (0.008) | 0.871 (0.003) | 0.908 (0.0) |
| | 27 | 0.822 (0.008) | 0.858 (0.001) | 0.813 (0.006) | 0.898 (0.003) |
| | full | 0.9 (0.001) | 0.891 (0.005) | 0.812 (0.006) | 0.795 (0.003) |
| LSTN | 1 | 0.871 (0.001) | 0.894 (0.0) | 0.86 (0.003) | 0.872 (0.004) |
| | 5 | 0.849 (0.006) | 0.885 (0.001) | 0.88 (0.0) | 0.909 (0.003) |
| | 16 | 0.875 (0.002) | 0.847 (0.008) | 0.83 (0.006) | 0.918 (0.001) |
| | 27 | 0.848 (0.006) | 0.893 (0.002) | 0.882 (0.0) | 0.903 (0.003) |
| | full | 0.888 (0.001) | 0.848 (0.004) | 0.905 (0.001) | 0.86 (0.003) |
| RNN | 1 | 0.861 (0.003) | 0.854 (0.002) | 0.806 (0.009) | 0.846 (0.003) |
| | 5 | 0.87 (0.003) | 0.841 (0.006) | 0.875 (0.001) | 0.824 (0.008) |
| | 16 | 0.843 (0.006) | 0.841 (0.007) | 0.819 (0.009) | 0.836 (0.005) |
| | 27 | 0.852 (0.004) | 0.903 (0.004) | 0.844 (0.006) | 0.891 (0.0) |
| | full | 0.82 (0.009) | 0.869 (0.002) | 0.884 (0.0) | 0.819 (0.009) |
| LSTM | 1 | 0.866 (0.001) | 0.843 (0.003) | 0.85 (0.003) | 0.872 (0.001) |
| | 5 | 0.872 (0.003) | 0.849 (0.006) | 0.882 (0.0) | 0.849 (0.004) |
| | 16 | 0.878 (0.002) | 0.858 (0.006) | 0.887 (0.001) | 0.888 (0.001) |
| | 27 | 0.846 (0.004) | 0.831 (0.007) | 0.859 (0.004) | 0.862 (0.004) |
| | full | 0.891 (0.001) | 0.864 (0.0) | 0.872 (0.003) | 0.877 (0.001) |
| CNN | 1 | — | — | — | — |
| | 5 | — | — | — | — |
| | 16 | — | — | — | — |
| | 27 | — | — | — | — |
| | full | 0.836 (0.003) | — | — | — |

Table 6: Mean episodic rewards under distribution shift in HalfCheetah environment normalized by performance of the expert policy under distribution shift. Rewards are given $\pm 1$ SE which is shown in parentheses.

## A.10 INITIALIZATION OF RECURRENT WEIGHTS

We leverage an initialization scheme consistent across each of the recurrent networks in order to control for potential confounding effects that could arise due to differences present in the default initializers for the recurrent weights. We will first discuss what the default initialization scheme looks like for the recurrent networks we considered and then motivate our proposed method of initialization.

**Default initialization schemes.** We define the default initialization schemes of recurrent weights in RNNs, LSTMs and GRUs as those implemented by TensorFlow (Abadi et al., 2016) and the default initialization of recurrent weights in a CfC as given by the open-source implementation presented in Hasani et al. (2021). The default initializers for these models differ in a couple key areas. For one, RNNs, LSTMs and GRUs are orthogonally initialized whereas CfCs are initialized from a Glorot uniform distribution.

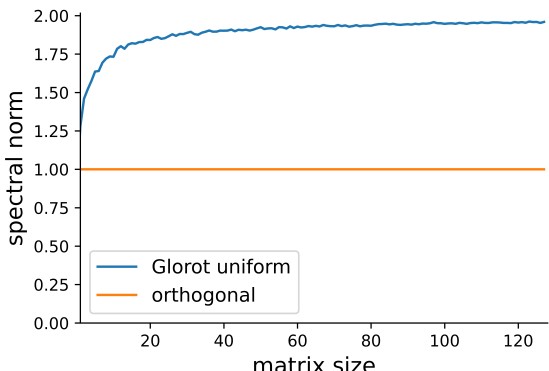

Figure 30: Spectral norm of random square matrices drawn from orthogonal and Glorot uniform distributions for various sized matrices.

In Figure 30, we investigate the spectral norm of random matrices drawn from orthogonal and Glorot uniform distributions. Due to the significant disparity between the spectral norms inherent to the distributions, we opt to standardize the initialization scheme across all recurrent models by drawing the weights from an orthogonal distribution. We motivate this further in the description of our initializer.

Another concern is that RNNs, LSTMs and GRUs are distinct with respect to how they are orthogonally initialized: in particular, consider $[\boldsymbol{W}_0], [\boldsymbol{W}_0, \boldsymbol{W}_1, \boldsymbol{W}_2, \boldsymbol{W}_3], [\boldsymbol{W}_0, \boldsymbol{W}_1, \boldsymbol{W}_2]$ which represent the concatenated recurrent weights in RNNs, LSTMs and GRUs respectively. Rather than initialize each $\boldsymbol{W}_i$ to be orthogonal, TensorFlow draws the concatenated set of matrices from an orthogonal distribution. The following result demonstrates why this is problematic in the context of our work.

**Lemma 1.** *Consider matrices $\boldsymbol{A} \in \mathbb{R}^{n \times p}$ and $\boldsymbol{B} \in \mathbb{R}^{n \times q}$. Let $\boldsymbol{C} = [\boldsymbol{A} \ \boldsymbol{B}]$ denote the concatenation of the two matrices. Then, $||\boldsymbol{A}|| \leq ||\boldsymbol{C}||$ and $||\boldsymbol{B}|| \leq ||\boldsymbol{C}||$ where $|| \cdot ||$ denotes the spectral norm.*

*Proof.* We will first consider the case of matrix $\boldsymbol{A}$. By definition, $||\boldsymbol{A}|| = \sup_{\boldsymbol{x} \neq 0} \frac{||\boldsymbol{A}\boldsymbol{x}||}{||\boldsymbol{x}||}$. If we constrain $\boldsymbol{x}$ to be of unit norm, then $||\boldsymbol{A}|| = \sup_{\boldsymbol{x} \neq 0} ||\boldsymbol{A}\boldsymbol{x}||$. Let $\boldsymbol{z}$ denote the unit norm vector that maximizes $||\boldsymbol{A}\boldsymbol{x}||$. Then, we can construct another vector $\boldsymbol{m} = [\boldsymbol{z} \ 0 \ 0 \cdots \ 0]$ which represents the concatenation of $\boldsymbol{z}$ which contains $p$ entries and a zero-vector which contains $q$ entries. Note that by construction $\boldsymbol{m}$ is also unit norm. It follows that $||\boldsymbol{A}\boldsymbol{z}|| \leq ||\boldsymbol{C}\boldsymbol{m}||$ which means that the maximum attainable value of $||\boldsymbol{C}\boldsymbol{x}||$ is greater than or equal to the maximum attainable value of $||\boldsymbol{A}\boldsymbol{x}||$ over all unit vectors $\boldsymbol{x}$. Thus, $||\boldsymbol{A}|| \leq ||\boldsymbol{C}||$. An analogous argument can be made to show that $||\boldsymbol{B}|| \leq ||\boldsymbol{C}||$. $\square$

By Lemma 1, it follows that the default initialization in LSTMs and GRUs produces submatrices $\boldsymbol{W}_i$ with spectral norm less than the spectral norm of the default initialized recurrent weights in RNNs. We will address this discrepancy in our proposed initialization scheme.

**Orthogonal spectral-initialization scheme.** Orthogonal spectral-initialization refers to the initialization scheme we utilized for the models in all the results we presented. First and foremost, this standardizes the distribution of the recurrent weights to be orthogonal. Note that we also formalize an analogous initializer which we call Glorot uniform spectral-initialization which is detailed in the next subsection.

In practice, we use the orthogonal distribution as it leans on the intuition of our connectivity prior in which we show empirically that networks initialized at lower spectral norms tends to converge to solutions with lower spectral norms as well. Because we care about the robustness of the network as measured by spectral norm at convergence and found that orthogonal weights have lower spectral norm than Glorot uniform ones (Figure 30), we opt to draw from the orthogonal distribution.

With respect to the concatenation issue discussed above, we address this by instead drawing each recurrent weight matrix from an orthogonal distribution as opposed to drawing the full, concatenated set of matrices from one. This ensures equivalent spectral norms across all recurrent weights and models.

Finally, the spectral-initialization refers specifically to the rank-r SVD performed on the full-rank weights $W_{rec}$ to generate $W_1, W_2$ (Section 3.1) for training. We opted to perform a spectral decomposition as opposed to drawing $W_1$ and $W_2$ from orthogonal distributions individually because $W_1 W_2$ is no longer orthogonally distributed. In addition, note that multiplying the two matrices does not preserve the spectral norm of $W_{rec}$. In contrast, by the Eckhart-Young-Minsky theorem, a rank-r SVD is the most efficient approximation of $W_{rec}$ under the spectral norm.

Note that the drawback of this initialization scheme is that we are unable to prove all the results posed in Theorem 1 due to the dependence between entries in an orthogonal matrix (appendix A.6.2). In contrast, we are able to prove Theorem 1 in the case of uniformly distributed matrices, which we motivate next.

**Glorot uniform spectral-initialization scheme.** The Glorot uniform spectral-initializer refers to the same construction as the orthogonal spectral-initializer, except recurrent weights are drawn from Glorot uniform distributions instead. Note that this distribution is valuable if one wants theoretical guarantees for all the properties given in Theorem 1. However, this comes at the cost of empirically less robust models as evidenced by the learned weights possessing higher spectral norm relative to their orthogonally initialized counterparts (results not shown).

## A.11 DETAILS OF EXPERIMENTAL SETUP

In this section, we extensively detail each of the experiments and analyses that were presented in the main portion of the paper.

### A.11.1 GENERATING EXPERT TRAJECTORIES

**Arcade learning environments.** Within the set of ALEs, we considered the Seaquest and Alien environments. These environments in particular were chosen as we were able to train reinforcement learning agents that achieved performance competitive to state-of-the-art using out-of-the-box models provided by RLlib (Liang et al., 2017). For the ALEs, we train an Ape-X DQN (Horgan et al., 2018b) model using the Atari pre-processing framework detailed in Horgan et al. (2018b). We employ the default implementation of the Ape-X DQN model given by RLlib which is a tuned version of the model presented in Horgan et al. (2018b). After the model is trained to perform sufficiently well in the environment, we use it to generate expert trajectories. Specifically, we generate 100 rollouts of the trained model's observations and actions taken in the environment which were later used to fit the recurrent models we examine in an imitation learning framework.

**MuJoCo.** Within the set of MuJoCo environments, we considered the HalfCheetah environment. Again, this environment was chosen based on its amenability to models provided by RLlib. For the MuJoCo environment, we employ an implementation of proximal policy optimization (PPO) given by RLlib (Schulman et al., 2017). We also leverage the default set of hyperparameters given by RLlib to train the model. As we did with the ALEs, we generate 100 rollouts of the trained model's observations and actions taken in the environment which were later used to fit the recurrent models using imitation learning.

### A.11.2 IMITATION LEARNING FRAMEWORK

For each recurrent architecture, we construct a model to be fit offline on the generated expert trajectories. We have layers in the network that act to preprocess the observations before entering the recurrent portion of the network as well as output layers from the recurrent portion in accordance to the action space specified by the environments. Since the observation and action spaces in ALEs and MuJuCo environments are distinct, we construct different network architectures for each.

First, we consider the network architectures used for ALEs. In their raw form, observations are given by a $210 \times 160 \times 3$ dimensional image, but they are preprocessed using the Atari preprocessing specified by Horgan et al. (2018b). This yields new observations of dimension $84 \times 84 \times 3$ which are used as inputs to the model.

| layer type | activation |
|---|---|
| Conv(64, k=5, s=2) | relu |
| Conv(128, k=5, s=2) | relu |
| Conv(128, k=5, s=2) | relu |
| AveragePooling | none |
| TimeDistributed | none |
| recurrentNet | none |
| FC(numActions) | softmax |

Table 7: ALE network architecture

In the table above, $k$ denotes the kernel size and $s$ denotes the stride of the convolution. recurrentNet refers to the recurrent network module that was varied and is discussed in further detail below. numActions refers to the number of actions the agent can take in the environment. Since the action space is a discrete categorical variable that can take on numActions different values, the network is trained using categorical cross-entropy loss. We detail the model hyperparameters and corresponding grid search in A.11.3.

Next, we will consider the network architecture employed for MuJoCo networks.

| layer type | activation |
|---|---|
| FC(256) | relu |
| FC(256) | relu |
| TimeDistributed | none |
| recurrentNet | none |
| FC(numActions) | tanh |

Table 8: MuJoCo network architecture

Since the action space for MuJuCo environments in continuous, the output is a continuous variable that can take on numActions different values and the network is trained using MSE loss. Furthermore, since the action space is bounded within the interval $[-1, 1]$, a tanh activation function is applied to the output layer.

### A.11.3 MODEL HYPERPARAMETERS

Below is a table with the model hyperparameters that were used in both the ALE networks and MuJoCo networks. Parameters in square brackets represent a grid search over which the best performing model was chosen. Note that only offline performance on the validation/test set was used to determine the best performing model in the grid search. Parameters in curly braces are dimensions across which the network was individually evaluated for comparison.

The architecture of the convolutional head in the ALE networks was used on the basis of work presented in Lechner et al. (2022) which conducting extensive grid searching across convolutional architectures over many ALEs. Analogously, the dense head used in the MuJoCo architecture was drawn from Hasani et al. (2021).

| hyperparameter | value |
|---|---|
| optimizer | Adam ($\beta_1 = 0.9$, $\beta_2 = 0.999$) |
| hidden size | 64 |
| learning rate | $[5 * 10^{-5}, 1 * 10^{-4}, 5 * 10^{-4}]$ |
| epochs | 150 |
| rank | $\{1, 5, 16, 27, \text{full}\}$ |
| sparsity | $\{0, 0.2, 0.5, 0.8\}$ |

Table 9: Model hyperparameters

In all the environments, we considered models of rank $1, 5, 16, 27, \text{full}$ and sparsities of $0, 0.2, 0.5, 0.8$. Note that the recurrent weights in the full rank networks are not rank decomposed, as they are for the low-rank networks. These ranks and sparsities were chosen such that for each rank we examined, we also looked at a sparsity level for which the number of recurrent parameters in each parameterization is roughly equivalent. For example, since the networks we constructed had a hidden dimension of size $64$, a recurrent matrix with a sparsity level of $0.5$ has the same number of parameters as one with a rank of $16$. We did not present results of networks with a very high recurrent sparsity like $0.95$ (which would roughly be the sparse analog to rank-1 networks) due to optimization difficulties encountered, particularly in LSTMs and GRUs. In order to ensure that the performance of a given network wasn't due to a favorable random sparse mask, we trained each model 3 times from different random seeds and averaged over the model performance.

Finally, we also employed an initialization scheme for the recurrent weights distinct from the default initialization schemes given in open-source implementations. For details on this, refer to A.10.

### A.11.4    EVALUATION METRICS

Trained models were evaluated offline on a validation/test set with respect to their cross-entropy loss in the case of ALE networks and mean-squared error in the case of MuJoCo networks. However, in practice, we care about how the agent performs when deployed in the environment closed-loop. The online rewards were computed by performing 10 simulations of the agent in the environments and averaging over the rewards from each episode. The in-distribution setting refers to the environments in which the observations are not modified aside from preprocessing performed by RLlib.

In the distribution shift setting, for the ALE observations we perform 5 types of perturbations to the agent's observations: cutout, brighten, darken, noise, and blur. In cutout, we remove some pixels from the center of the input image. In brighten, we shift all the pixel values by a constant amount in the positive direction and do the same for darken but shift the pixels in the negative direction. To noise the image, we draw random noise from a uniform distribution and add it to the input image. For distribution shifts in the MuJoCo environments, we perform 3 types of perturbations: noise, dropout and offset. To noise the observations, we draw random noise from a normal distribution and add it to the observation vector. To blur the observations, we perform a Gaussian blurring on the image. To perform dropout, we mask a subset of the dimensions in the observation vector. To perform offset, we, with equal probability, shift the observation vector in the positive or negative direction by an amount constant across all dimensions. For each of the inputs during closed-loop navigation, we apply the distribution shift with probability $p = 0.1$.

To evaluate the models, both in-distribution and under distribution shift, we normalize the rewards by the performance of the expert policy in order to convert the rewards into units that allow us to benchmark the performance of the imitation learning agents while also making comparisons across recurrent models and connectivities.

### A.11.5    DETAILS ON NETWORK ANALYSES

In this section, we further elaborate on the chosen modes of analysis in the trained recurrent networks and in cases where the metric computation is potentially ambiguous we elaborate on the methodology used to compute it. All the plots shown in the paper regarding these analyses were performed on models trained in the Seaquest environment.

**Spectral analyses.** For recurrent networks that have more than one set of recurrent weights (CfCs, LSTMs, GRUs), any of the spectral statistics that were computed (i.e. eigenspectrum, recurrent weights singular value spectrum, input weights singular value spectrum, etc.) we compute and plot a single statistic that represents the average over the statistics of each of the individual recurrent weight matrices.

On a separate note, with respect to measuring the decay of the singular value spectrum, note that we normalized the sorted set of singular values by the spectral norm. This was done in order to explain away the magnitude of the transformation, since we separately motivate and analyze the spectral norm. Doing so also ensures each of the decay curves start at 1 which enables comparison across models, ranks and sparsities.

**Recurrent gradient analysis.** In Section 4.2, we briefly discussed the analysis performed to analyze the evolution of the recurrent gradients across time. Here, we provide some additional details that were not addressed in the main portion of the paper.

Firstly, we computed the norm of the gradients in log space due to the exponential decay associated with the gradients over time (which happens in practice because gradient propagation is multiplicative). Secondly, note that each of the recurrent gradients start evolving from $0$. We enforced the decay to start from $0$ since we do not care about the starting value of the gradients at the end of time, but rather how this gradient evolves in units relative to the start value (i.e. the rate of decay matters, but not the value it started decaying at). So, we translated the curves to begin at $0$, irrespective of their starting value.

**Effective dimensionality of trajectories.** To understand the complexity of the dynamics across recurrent models, connectivity ranks and connectivity sparsities, we collected the trajectories of the model during the simulation of the agent in the online, closed-loop setting and fit a PCA on the entire set of trajectories. We then computed the explained variance of the top 5 principal components (PCs) as a proxy for the effective dimensionality of the trajectories. This is a canonical approach to analyzing state-space trajectories in recurrent networks (Lechner et al., 2020).

However, we extended the canonical analysis beyond just the full state-space trajectories by decomposing the trajectories into recurrently-driven activity and input-driven activity. In particular, we considered $W_{rec}h_{t-1}$ as the recurrent activity and $W_{inp}x_t$ as the input activity. We analogously aggregated these trajectories individually and computed the explained variance of the top 5 PCs. This gives us separate measurements for what the dimensionality of activity looks like in the recurrent subspace versus the input subspace and allows us to explicitly disentangle these two axes.

For recurrent networks that have more than one set of recurrent (and input) weights (CfCs, LSTMs, GRUs), we computed the explained variance of the top 5 PCs individually for each set of weights and then averaged across them to produce a single number summary for the complexity of network dynamics.

## A.12 MODEL PARAMETER COUNTS

| model | parameter count |
|-------|-----------------|
| CfC | 61632 |
| LSTM | 81726 |
| RNN | 20544 |
| GRU | 61632 |

Table 10: Parameter counts of the fully-connected, full-rank versions of each recurrent model. Input size = 256, hidden state size = 64.

