# OpenReview forum: "Leveraging Low-Rank and Sparse Recurrent Connectivity for Robust Closed-Loop Control"
_ICLR.cc/2024/Conference — ICLR 2024 spotlight_

### Official Review · Reviewer_R5dM · 2023-10-18

**Soundness:** 4 excellent
**Presentation:** 4 excellent
**Contribution:** 3 good
**Rating:** 8
**Confidence:** 3

**Summary:**

Low-rank and sparse initial recurrent connectivity have been found to be two powerful priors when training RNNs to imitate an expert and later deploy them (in control tasks). This work aims to better understand the reasons behind this behavior. From a theoretical point of view, it links sparsity levels and rank at initialization with properties of the recurrent connectivity matrix. Empirically, it studies in depth how those two properties affect the dynamics of the different networks and analyzes how they affect generalization to distribution shifts.

**Strengths:**

The paper is very well written. Theoretical results are simple and insightful for the result of the result. The empirical analysis is thoroughly done. It nicely combines analysis tools developed in dynamical systems, computational neuroscience, and deep learning to better understand the observed behavior.

**Weaknesses:**

The analysis focuses on the spectral properties of the connectivity matrix. While I can understand why this is a reasonable choice for understanding the impact on low-rank and sparsity for a single architecture, it seems limited when it comes to comparing different architectures. Ideally, one would need to study the spectral properties of the recurrent Jacobian. The recurrent Jacobian would heavily depend on the recurrence connectivity matrix for all architectures, but important differences might still remain. The paper is currently missing a discussion of this point.

Minor:

- It would be great to know which kind of distribution shift you are using in your experiments. I could not find this information.
- the work of Herbert Jaeger or Wolfgang Maass is probably a better reference for echo state networks than the Deep Learning Book of Goodfellow et al.

**Questions:**

c.f. Weaknesses

---

> ### Author Response · Authors · 2023-11-21
> **Response to reviewer R5dM**
>
> We thank the reviewer for the largely positive review of our work! We address your comments below:
>
> 1. The analysis focuses on the spectral properties of the connectivity matrix. While I can understand why this is a reasonable choice for understanding the impact on low-rank and sparsity for a single architecture, it seems limited when it comes to comparing different architectures. Ideally, one would need to study the spectral properties of the recurrent Jacobian. The recurrent Jacobian would heavily depend on the recurrence connectivity matrix for all architectures, but important differences might still remain. The paper is currently missing a discussion of this point.
>
> We completely agree and note that Reviewer 59d7 shared a similar sentiment. To address this, we perform an analysis on the recurrent Jacobians with can be found in Section 4.2 in which we compute the norm of the gradients backwards through time in order to understand how the network’s recurrent attention evolves as a function of the amount of time that has passed since it observed a given input. We are able to show that the spectral radius is indeed a viable proxy for the rate at which the gradient evolves across time across models, recurrent ranks and recurrent sparsities, which is reassuring for the strength of the proposed parameterization as a modulator of recurrent memory.
>
> 2. It would be great to know which kind of distribution shift you are using in your experiments. I could not find this information.
>
> This information can be found in Appendix A.11 in the evaluation metrics section. We employ canonical distribution shifts such as adding random noise and brightening/darkening/blurring input images.
>
> 3. the work of Herbert Jaeger or Wolfgang Maass is probably a better reference for echo state networks than the Deep Learning Book of Goodfellow et al.
>
> Agreed, this has been corrected.
>
> Thanks again for the positive evaluation of our work and we hope the recurrent Jacobian analysis has alleviated any potential concerns about the viability of the spectral analyses as proxies for network dynamics.

---

> > ### Comment · Reviewer_R5dM · 2023-11-22
> >
> > Thanks to the authors for clarifying those points. I keep my score as it is.

---

### Official Review · Reviewer_XFJy · 2023-10-30

**Soundness:** 2 fair
**Presentation:** 2 fair
**Contribution:** 2 fair
**Rating:** 6
**Confidence:** 2

**Summary:**

The paper investigates the robustness of recurrent-based agents on non-stationary environments. In particular, the paper focuses on how sparsity and rank of different recurrent neural networks (RNNs) impact network dynamics. The paper provide theoretical insight on the property of the weight matrices in RNNs with respect to the rank and sparsity, which is used to support their empirical findings. Specifically, by considering the spectral radius and the spectral norm of the weight matrices, the paper suggests that smaller spectral radius and norm can yield better generalization under distribution shift under their experiments.

**Strengths:**

- The paper does a thorough investigation on how the sparsity and rank of the weight matrices impact the generalization of an agent under distribution shift.
	- The paper focuses on many axes in the experimentation.

**Weaknesses:**

**Comments**
Addressing the following comments will increase my score:
- The paper is very difficult to read---while the paper provides some high-level intuition for particular quantities (e.g. low sparsity corresponding to the rate of vanishing gradient), I feel there is a big gap between every subsection. In other words, I often have trouble making the connections between the results. Furthermore, even with the large text of paragraphs, often the most important information is ambiguous, and I list them in the following:
	- I don't think I completely understand this experiment. What is $W_{full}$? I expected the full state-space trajectories mean running PCA on the states gathered over time, and the two former corresponding to $W_{rec}h_t, W_{inp}x_t$.
	- Theorem 4.1 seems to relate the weight matrices based on particular initialization schemes. I understand that in figure 4 the paper appears to agree with theorem 4.1, but I fail to understand why this theorem needs to be in the main paper as it does not explain any insight on the generalization/robustness directly.
	- The scale of the subplots should be consistent for easier comparison (e.g. figures 4, 5, and 6.)
- The proof of theorem 4.1 is unclear to me. While I see that the paper uses random matrix theory to obtain result on the spectral radius and spectral norm based on the sparsity, I don't completely understand how we obtain the statements for rank $r$ for both initializations.
- Regarding the reward ranking metric: Why not normalized return? Ranking is not too meaningful regarding how generalized the agent is---it is only relative performance. An alternative is to provide the expert performance as a baseline.
- On figure 5, the paper only indicates the variance of the data captured by the top 5 principal components (PCs), we lose the information on the remaining variance captured by the proceeding PCs. As a result, using only the top 5 PCs to claim higher/lower effective dimensionality seems incorrect. For example, comparing two methods $M_1$ and $M_2$, if we assume that $M_1$'s top 5 PCs capture 0.7 explained variance while $M_2$'s capture 0.5, we cannot guarantee that $M_1$ is always capturing more variance as we increase the number of PCs.

**Questions:**

See comments above.

---

> ### Author Response · Authors · 2023-11-21
> **Response to reviewer XFJy**
>
> We thank the reviewer for your review of our work. We have incorporated many of your suggestions into the revised draft and respond to your specific points below.
>
> 1. The paper is very difficult to read---while the paper provides some high-level intuition for particular quantities (e.g. low sparsity corresponding to the rate of vanishing gradient), I feel there is a big gap between every subsection. In other words, I often have trouble making the connections between the results.
>
> We tend to agree with the reviewer that the previous draft of the paper was difficult to parse given the gaps between sections. We have refactored the paper structure and rewritten sentences such that the flow of the story is more coherent and each of the analyses are more thoroughly motivated. Please refer to the comment addressed to all the reviewers to see both the structural and experimental changes we have made.
>
> 2. Furthermore, even with the large text of paragraphs, often the most important information is ambiguous, and I list them in the following:
>
> * I don't think I completely understand this experiment. What is $W_{full}$? I expected the full state-space trajectories mean running PCA on the states gathered over time, and the two former corresponding to $W_{rec} h_t$ and $W_{inp} x_t$.
> * Theorem 4.1 seems to relate the weight matrices based on particular initialization schemes. I understand that in figure 4 the paper appears to agree with theorem 4.1, but I fail to understand why this theorem needs to be in the main paper as it does not explain any insight on the generalization/robustness directly.
> * The scale of the subplots should be consistent for easier comparison (e.g. figures 4, 5, and 6.)
>
> We respond to each in turn:
>
> * Yes, your interpretation of the individual state-spaces is correct and we clarified this in the paper. The purpose of this experiment is to disentangle the recurrently-driven and input-driven portions of the state-space. We show that changes in the recurrent connectivity primarily affect the recurrently-driven portion of the subspace in the in-distribution setting and also that CfCs have the lowest-dimensional input state-space trajectories of the examined models. A canonical analysis which only examines the full-state space would not observe the impact of changes in the recurrent connectivity on the recurrently-driven subspace given that the full state-space trajectory appears to be dominated by the inputs in-distribution. We also added an analysis examining the state-space trajectories under distribution shift instead which can be found in A.1. There we show that unlike in-distribution, under distribution shift modulations in the recurrent connectivity actually affect the input and full state-space trajectories.
>
> * Theorem 1 proves properties (for some subset of cases presented in the paper) regarding how the spectral norm and radius of the recurrent weights change as a function of rank and sparsity. We motivate the relevance of these with respect to robustness more thoroughly in Sections 3 and A.5.2 but summarize them here. Spectral norm measures the magnitude of the expansion induced by a weight matrix and thus measured the extent to which a perturbation effects the hidden state (or input state in the case of the input weight matrix). Spectral radius acts as a proxy for the network's attention span across time and is thus a measure for how far into the future a perturbation will affect the system. Since we can modulate both via rank and sparsity, we maintain that this Theorem 1 is pertinent to the paper as it gives us provability regarding the efficacy of the parameterization.
>
> * The scale of the subplots is fixed in Figure 4a. In Figure 4b, we use the same scale horizontally but different scales vertically because we want the reader to be able to visualize the changes across ranks and sparsities in LSTMs (as well as see the SE bars) which would otherwise be squashed if we placed it on the same scale as the CfCs. Analogous logic was used for making the scales for Figure 5b. However, if the reviewer maintains that readability would be improved by standardizing the scales even in these cases, we are happy to make the modification.
>
> 2. The proof of theorem 4.1 is unclear to me. While I see that the paper uses random matrix theory to obtain result on the spectral radius and spectral norm based on the sparsity, I don't completely understand how we obtain the statements for rank
>  for both initializations.
>
> Apologies, this was a typo in the theorem statement that has been corrected. You are correct, we do not prove the trend regarding the spectral radius as a function of rank in the Glorot uniform case and have instead added an empirical analysis to A.6.

---

> ### Author Response · Authors · 2023-11-21
> **Response to reviewer XFJy continued**
>
> 3. Regarding the reward ranking metric: Why not normalized return? Ranking is not too meaningful regarding how generalized the agent is---it is only relative performance. An alternative is to provide the expert performance as a baseline.
>
> Agreed, this is a better metric for rewards. We have modified Figure 1 and the other results to normalize the rewards by the performance of the expert, as suggested.
>
> 4. On figure 5, the paper only indicates the variance of the data captured by the top 5 principal components (PCs), we lose the information on the remaining variance captured by the proceeding PCs. As a result, using only the top 5 PCs to claim higher/lower effective dimensionality seems incorrect.
>
> Using the top 5 PCs in particular was an arbitrary choice in order to analyze dimensionality. We agree that in order to be precise one should analyze the full set of PCs. We added a plot of this to Appendix A.8 (Figure 26) which shows that the conclusions made using only 5 PCs still hold: namely that CfCs have state space trajectories with lower dimensionality than LSTMs and RNNs.
>
> Again, thank you for your thorough attention to detail when reviewing our work, and we hope that our revisions resolve your concerns with the readability of the paper.

---

> > ### Comment · Reviewer_XFJy · 2023-11-21
> >
> > Thank you for the detailed feedback. I want to first note that the appendix is missing from the manuscript. Otherwise, thank you for restructuring the writing. I will increase the score after reading the new appendix.

---

> > > ### Author Response · Authors · 2023-11-21
> > > **Clarifying paper submission**
> > >
> > > Apologies for the confusion. We have placed the appendix in the supplementary material due to size constraints encountered when uploading the file. We are currently working on compressing the file in order to merge the main paper and appendix into a single file and will upload that as soon as we are done but we hope this isn't too much of an inconvenience for now.

---

> ### Author Response · Authors · 2023-11-22
> **Appendix has been added back into main submission**
>
> We have revised the manuscript such that the Appendix can now be found in the main submission, which we hope improves readability. Thank you for your patience!

---

> > ### Comment · Reviewer_XFJy · 2023-11-22
> >
> > Thank you for the update. I believe you have mostly resolved my previous questions.
> >
> > Thank you for the discussions but I have few more questions regarding the setting and potential impact of the paper after reading the appendix further:
> > 1. I might have misunderstood what distribution shift meant previously. On page 19, the paper says "distribution shifts are applied to the input image, which is then fed through a set of convolutional layers before entering the recurrent portion of the network." Is this distribution shift fixed? Changing over time? Does non-stationarity from policy change during online learning play into this distribution shift? Can the authors please clarify this?
> >   1. There is also both RL and IL (in appendix), how does this play into the distribution shift setting?
> > 1. Does figure 27(c) correspond to the offline performance? Why do we not use reward after offline training as a metric?
> > 1. As the theorem is actually a bit more restrictive, I am curious what the authors think about the remaining gaps (i.e. particular initialization not following a particular trend specified in the bullet points)---in particular is there follow-up work that aims to address this theoretically, or does it generally not hold?

---

> ### Author Response · Authors · 2023-11-22
> **Response to reviewer XFJy**
>
> Thank you for your response! We respond to your points below.
>
> 1. I might have misunderstood what distribution shift meant previously. On page 19, the paper says "distribution shifts are applied to the input image, which is then fed through a set of convolutional layers before entering the recurrent portion of the network." Is this distribution shift fixed? Changing over time? Does non-stationarity from policy change during online learning play into this distribution shift? Can the authors please clarify this?
>
> Under distribution shift, we refer to the neural policy being run in environments that differ in one aspect of their dynamics or observation from the environment with which the policy was trained with. For example, the presence of noise, omission of parts of the observation etc. The distribution we sample from to perturb the inputs is fixed across time. We will clarify this better in the paper.
>
> 2. There is also both RL and IL (in appendix), how does this play into the distribution shift setting?
>
> The models were trained using IL (for the data collection process RL was used). There are two reasons why we choose this approach. First, we wanted to replicate the imitation learning paradigm found in training neural agents in practice where no accurate simulation environment is available (autonomous driving, language agents, etc). Second, IL allows us to run fairer evaluations due to RL being inherently stochastic (sampling).
>
> 3. Does figure 27(c) correspond to the offline performance? Why do we not use reward after offline training as a metric?
>
> Figure 27c refers to the loss of the models trained by imitation learning evaluated on a validation set (so cross-entropy loss in the case of the Seaquest environment). In practice, we do not particularly care about the performance of the models evaluated on an offline, validation set but include them for the sake of detail. This is because the motivation of our work is the online, closed-loop setting in which evaluating the models via their online rewards is the canonical metric.
>
> 4.  As the theorem is actually a bit more restrictive, I am curious what the authors think about the remaining gaps (i.e. particular initialization not following a particular trend specified in the bullet points)---in particular is there follow-up work that aims to address this theoretically, or does it generally not hold?
>
> We note that the gaps in Theorem 1 corresponding to the cases we do not prove are empirically analyzed in Appendix A.6. In particular, **the claims that we do not prove still have the same trends as the relationships proved in the theoretical points**. To clarify there are 8 cases we aim to address, each of which can be delineated using by (initialization, spectral property, pruning method). We note that the cases we prove in the theorem are as follows: (uniform, norm, rank), (orthogonal, norm, rank), (uniform, norm, sparsity), (uniform, radius, sparsity), (orthogonal, radius rank).
>
> This leaves the following 3 cases we have not shown proof for: (orthogonal, norm, sparsity), (orthogonal, radius, sparsity), (uniform, radius, rank). We reiterate what the empirical analyses show in each of these cases.
>
> * Empirically, we show that the spectral radius increases as a function of rank in the Glorot uniform case (which is the same as the relationship between rank and spectral radius in the orthogonal case)
> * Empirically, we show that the spectral radius decreases as a function of sparsity in the orthogonal case (which is the same as the relationship between sparsity and spectral radius in the uniform case)
> * Empirically, we show that the spectral norm first increases and then decreases as a function of sparsity in the orthogonal case (which is similar to the relationship between spectral norm and sparsity in the Glorot uniform case, although the relationship there is monotonically decreasing)
>
> The reason it is difficult to prove the (orthogonal, sparse) case is that many results from random matrix theory rely on iid entries which don't hold for orthogonal matrices. Nonetheless, we make an attempt at imposing theory in the orthogonal, sparse case by assuming the entries are approximately Gaussian and provide a provable relationship under that assumption (which agreed reasonably well with the empirical trends given the plots in A.6). We note that improving upon the theoretical argument in this case remains an open challenge. And while we do not have proof for these cases at the moment, the fact that the empirical trends align with the theoretically motivated cases is reassuring. **Most importantly, in both uniform and orthogonal initialization, either a theoretical or empirical analysis supports both of the following: spectral radius and spectral norm decrease in sparsity and increase in rank.** This is precisely the knowledge we want from the perspective of being able to use the prior as a modulator of these spectral properties.

---

> > ### Comment · Reviewer_XFJy · 2023-11-22
> >
> > Thank you for the detailed response, I will increase the score to 6. My understanding now is that the sparsity/rank can help select the important features, thus is able to be more robust to these distribution shifts, is that fair?

---

> > > ### Author Response · Authors · 2023-11-22
> > > **Response to reviewer XFJy**
> > >
> > > Thank you for your response and score increase! We appreciate your engagement with us.
> > >
> > > Regarding your question, we would say that this is a fair characterization of the low-rank models, although the sparse connectivity is a bit more nuanced. In particular, we showed that increasing sparsity and decreasing rank are distinct with respect to the rate of decay they induce in the singular value spectrum of the recurrent weights. Namely, the former causes the decay rate to decrease and the latter causing the decay rate to increase.
> > >
> > > The decay rate of the spectrum acts as a proxy for the effective number of dimensions that recurrent weights expand the hidden vector in. If the decay rate is lower, then the recurrent weights place a more uniform expansion across all directions and if it is higher, they only transform the vector in a few notable directions.
> > >
> > > This effect is observable when analyzing the dimensionality of the recurrent state-spaces, in which we find that sparse models have higher dimensional trajectories whereas low-rank models have lower dimensional trajectories. And this is why we find that lowering the rank of the models is significantly more effective at encouraging robust models than increasing sparsity. **This points to the distinction between pruning by sparsity and pruning by rank at initialization**, one of the key findings of our work.

---

> > > > ### Comment · Reviewer_XFJy · 2023-11-22
> > > >
> > > > I see, thank you for the explanation on the difference between the two and the key findings.

---

### Official Review · Reviewer_59d7 · 2023-10-31

**Soundness:** 3 good
**Presentation:** 3 good
**Contribution:** 2 fair
**Rating:** 6
**Confidence:** 4

**Summary:**

This paper show how the spectral radius and norm depend on sparsity and the rank of the recurrent weights, using these as a proxy to robustness and to introduce an inductive bias towards better performance for networks trained in a causality gap setting.
Better performance under distribution shifts is demonstrated for low-rank and sparse recurrent neural networks for networks trained on various environments under an imitation learning framework.
Finally, it is shown that closed-form continuous-time neural networks are more amenable to a low-rank and sparse connectivity prior than the canonical recurrent architectures (vanilla RNNs and LSTMs).

**Strengths:**

Application of existing analyses to understand robustness and generalization performance to a new connectivity framework is novel. Furthermore, the analysis of  decomposing the activity into recurrent and input driven components seems novel.  The proof of the theorems are sound. All methods are well described in a way that allows for reproducible, especially if code will be made available. The main aim, to better understand why why sparse connectivity is useful in closed-loop systems is a very important problem to tackle.
The theoretical contributions are significant to an even broader setting than is considered in the paper.

**Weaknesses:**

Already existing analysis and tasks.
A more comprehensive comparison to other networks trained on the tasks is missing.

Some existing analyses to measure robustness, for example as the loss in performance as a function of noise level is missing.
Overall, it is not entirely clear why some of the proxies used in the paper are sufficient to be used to asses robustness.
The proxies are only partially justified to be good measures of robustness. Further analyses of the dynamics, in greater detail than dimensionality, would be important to have certainty about the robustness properties of the trained networks.
There are many other measures for assessing the effect of perturbations on dynamics, for example Lyapunov exponents and these should be calculated for a more complete analysis.

Because there is no assessment of the learning dynamics itself or the process of learning itself the vanishing gradient analysis is a misnomer. I understand the main point of these analyses to be some memory component or transient (convergent/divergent) dynamics itself instead of it being related to the question whether the network architecture can support vanishing gradients. That said, also for vanishing gradients Lyapunov exponents are a good method for analysis.

It is unclear why a particular number of samples were considered to be sufficient. For example, why are 3 randoms sufficient to rule out that a favorable random sparse mask is not influencing the results?
Furthermore, for such high spaces for the task, averaging over 5 perturbations seems insufficient.

Theorem 1 lacks the case where the recurrent matrix does not have full connectivity to begin with and the influence of the sparsity parameter $s$ on the spectrum in that case.

\paragraph{Reward}
It is difficult to asses to what (good) performance the reward range in Figure 3 corresponds to. Would it be possible to show what kind of actions the highest and lowest performing networks correspond to at least? The performance of the agents is a very relative concept for which a number is not sufficient to understand what the networks actually are doing when they perform the task (supposedly well). What is the maximum number of rewards?

\paragraph{Reward ranking}
It seems that the reward ranking is a difficult to asses proxy for performance on the perturbed environments. For a full picture it would be better to show what amount of reward decrease the different shift cause. How do the distribution shift change the reward range for example? If all networks perform badly, but the Cfc slightly better, can we still claim that they are robust?
The best performing networks on the online performance measure seem not to be performing particularly well on distribution shifted versions of the task. How does the ranking look like if the performance of the in-distribution reward is also taken into account?
Finally, to see what kind of perturbation is most damaging for the different networks would be very insightful. Performance per perturbation type could also show what kind of robustness the different networks display.

For showing these ranking, it would be perhaps better to show the actual (average) rank as a number instead of a color coded version.

**Questions:**

About the eigenspectrum analysis.
Why is it the case that the closer the distribution of eigenvalues is to uniform, the more balanced the attention profile of the recurrent weights is across dimensions of the eigen-transformed?
And why does this have implications for the dimensionality of recurrent state-space dynamics?


In Figure 5, 6, 7 etc, what does the colored region mean? Is it the variance? Is it a confidence interval? If there are only three networks per parameter setting shouldn't they all be shown?

Is it really counterintuitive that with increasing sparsity, the dimensionality of the recurrent trajectories increases? Isn't sparsity just functioning as effectively increasing the rank?

It is not clear why higher dimensional dynamics would lead to more robust networks, as claimed in Figure 16. This seems to be in disagreement with the claims about constraining the network to be low-rank improved robustness in Section 4.3.

How is $\Delta W$ calculated in Figure 7? If it is just the change of the parameter across training, the caption should mention that instead of saying that the change in weights \emph{during} training is shown.

What number of parameters do the different networks have? In particular, how many more parameters do Cfcs have as a result of having two vanilla RNNs in them and another gating mechanism? Be more clear about how $F$ is parameterized. How could the increase in the number of parameters explain the increase in performance?


Shouldn't a higher spectral norm contribute to higher dimensional dynamics? For a low spectral norm the network would quickly collapse onto a low-dimensional manifold. How do you explain then the RNNs have higher spectral norm and lower dimensionality in their dynamics?

The last claim of Theorem 1 is only proved for orthogonal initialization?

What are the parameters used for the Adam optimizer? The default ones? Mention.

---

> ### Author Response · Authors · 2023-11-21
> **Response to reviewer 59d7**
>
> We thank the reviewer for the thorough review of our work. We have incorporated many of your suggestions into the revised draft and believe that it has substantially improved the paper. For a summarization of the modifications we have made to the paper, please refer to the comment addressed to all of the reviewers. We respond to your specific points below.
>
> 1. Already existing analysis and tasks. A more comprehensive comparison to other networks trained on the tasks is missing.
>
> We agree that the set of tasks is standard, however we do not intend to extend the task setting in the scope of this work beyond the closed-loop setting in which the ALE and Mujoco environments are canonical benchmarks. Furthermore, we believe that the strength of our work lies in the extensive analyses we perform on the network and thus makes up for any perceived limitations regarding the breadth of tasks explored.
>
> We disagree that the analyses presented in this work are unoriginal. As you noted, the decomposition of the state-space trajectory is not a canonical approach. Additionally, analyzing the singular value decay (which we agree was not properly emphasized in the previous draft, and was also an analysis on the eigenvalues previously which is less pertinent and has been modified) is also not a common mode of analysis, and is what we use to distinguish between pruning by inducing sparsity and pruning by reducing rank.
>
> We note in the related works “Alternative parameterization of RNNs” subsection as to why we restrict the scope of our analysis to this set of networks.
>
> 2. Some existing analyses to measure robustness, for example as the loss in performance as a function of noise level, is missing. Overall, it is not entirely clear why some of the proxies used in the paper are sufficient to be used to assess robustness. Further analyses of the dynamics, in greater detail than dimensionality, would be important to have certainty about the robustness properties of the networks.
>
> We examined 5 different, canonical distribution shifts and aggregated across them in order to measure robustness, (e.g., additive noise, blurring, darkening, etc.). Doing this as a function of the noise is computationally expensive, given that we train 20 sub-models for each architecture. Furthermore, our results comparing model performance are statistically significant, as can be verified in the tables in Appendix A.8.
>
> We more thoroughly motivate the chosen proxies for robustness in Section 3 and detail them in depth in Appendix A.5.2. We summarize how we motivate them in the paper below.
>
> * We motivate the analysis of the spectral norms more concretely as a measure of the magnitude of the transformation induced by the weights on a perturbation applied to the input (and thus also on the hidden state). We have also added in an analysis on the input spectral norms (as opposed to only the recurrent one) since the combination of the two better captures the robustness of the network. We also motivate the spectral norms via their relationship to the Lipschitz constant of the network (which is a canonical measure of robustness), but acknowledge the looseness of this approximation in the paper.
>
> * We motivate the decay of the singular value spectrum as a proxy for the effective number of directions the perturbation is transformed in. This is important since while the spectral norm captures a worst-case perspective of robustness, in practice also we care about how much the perturbation is expanded in each direction because if we imagine sampling distribution shifts from a uniform distribution, all directions are equally important.
>
> * The dimensionality analysis serves to capture this notion of “effective number of directions transformed in” concretely by looking at the representations of the data across time via the state-space trajectory analysis. In particular, the higher dimensional the trajectory is, the more directions the state-space dynamics evolve in and hence the more directions the perturbation affects, which yields poor robustness. This has also been clarified in the paper.
>
> 3. That said, also for vanishing gradients Lyapunov exponents are a good method for analysis.
>
> We agree with the spirit of the Lyapunov exponent analysis in that making claims about the recurrent memory using only the spectral radius is not sufficient; rather, we need to analyze the recurrent Jacobians themselves to account for the contribution of the hidden state to the gradient (similar sentiment shared by R5dM). However, we note that Lyapunov exponents are computed in the infinite time limit (Vogt et. al 2020) and for our purposes we care more so about the finite-time horizon specified by the task. Thus, we instead computed the norm of the recurrent gradient as a function of time and added this analysis to the paper (Figure 4b). We find that this analysis aligns with the spectral radius (Figure 4a), which is reassuring of its viability as a proxy for recurrent memory.

---

> ### Author Response · Authors · 2023-11-21
> **Response to reviewer 59d7 continued**
>
> 4. There are many other measures for assessing the effect of perturbations on dynamics, for example Lyapunov exponents and these should be calculated for a more complete analysis.
>
> As you stated, the Lyapunov exponent is a measure for both the recurrent memory and robustness. However, we would like to clarify that this measure of robustness is across the time dimension. In particular, it measures the effect of a perturbation applied to the hidden state into the future. We agree that this is a relevant measure of robustness and have clarified this in Section 4.2 when analyzing the recurrent memory via the recurrent Jacobian analysis (which is effectively capturing the same thing as the Lyapunov exponents). In contrast, the spectral norm and dimensionality analysis measure robustness in the input and hidden state representations at a given point in time. In particular, if we perturb the input $x_t$, the input weights spectral norm measures the worst-case expansion on the perturbation induced by $W_{inp}$ (and analogously for $h_t $ and the spectral norm of $W_{rec}$) which characterizes the magnitude change in the latent representation caused by the perturbation. The dimensionality analysis characterize the effective number of directions the perturbations can affect. We describe the intuition here in further detail in A.5.2.
>
> 5. Because there is no assessment of the learning dynamics itself or the process of learning itself, the vanishing gradient analysis is a misnomer. I understand the main point of these analyses to be some memory component or transient (convergent/divergent) dynamics itself, instead of it being related to whether the network architecture can support vanishing gradients.
>
> Agreed, we have renamed mentions of the vanishing gradient to recurrent memory horizon and instead motivate the analysis from the perspective of characterizing the attention profile of the network across time.
>
> 6. It is unclear why a particular number of samples were considered to be sufficient. For example, why are 3 randoms sufficient to rule out that a favorable random sparse mask is not influencing the results? Furthermore, for such high spaces for the task, averaging over 5 perturbations seems insufficient.
>
> We determined 3 random masks to be sufficient given the stability in the spectral statistics, as noted by the standard error in all the plots (given by the shaded area). Regarding the perturbations, we considered a breadth of possible distribution shifts (i.e. noise, brightening, blurring, etc.) and note that within each model, we consider 20 sub models or (rank, sparsity) pairs. Given the fact that trends hold amongst groups of (rank, sparsity) pairs and not just single models, we maintain that our analysis is sufficiently exhaustive.
>
> 7. Theorem 1 lacks the case where the recurrent matrix does not have full connectivity to begin with, and the influence of the sparsity parameter on the spectrum in that case.
>
> Sorry, we are unclear what the reviewer means by this question. Theorem 1 analyzes the spectrum as a function of rank in matrices with no sparsity and as a function of sparsity in matrices that are full-rank.
>
> 8. \paragraph{Reward} It is difficult to assess to what (good) performance the reward range in Figure 3 corresponds to. Would it be possible to show what kind of actions the highest and lowest performing networks correspond to at least? The performance of the agents is a very relative concept for which a number is not sufficient to understand what the networks actually are doing when they perform the task (supposedly well). What is the maximum number of rewards?
>
> We agree with the points you have brought up here, and have thus modified the reward results by normalizing by the performance of each of the models by the performance of the expert policy. These can be found in Figure 1. We find that the top performing imitation learning agents perform as well and in some cases slightly exceed the performance of the expert in-distribution.
>
> 9. \paragraph{Reward ranking} It seems that the reward ranking is a difficult to asses proxy for performance on the perturbed environments. For a full picture it would be better to show what amount of reward decrease the different shift cause. How do the distribution shift change the reward range for example? If all networks perform badly, but the Cfc slightly better, can we still claim that they are robust?
>
> We also agree and now we normalize the performance of the models under distribution by the performance of the expert in the distribution-shifted environment and then take the average of the distribution shifts in this normalized space. These can be found in Figure 1. We find that the most robust imitation learning agents are more robust than the expert policy, which points to the importance of recurrent supervision as well as constructing a robust recurrent state in the distribution shift setting.

---

> > ### Author Response · Authors · 2023-11-21
> > **Response to reviewer 59d7 continued**
> >
> > 10. The best performing networks on the online performance measure seem not to be performing particularly well on distribution shifted versions of the task. How does the ranking look like if the performance of the in-distribution reward is also taken into account?
> >
> > * Yes, this is due to the disparity between pruning by inducing sparsity and pruning by reducing rank, which is one of the major points we discover in this work
> > * A ranking that takes into account both in-distribution and distribution-shift performance is entirely a function of how much weight is placed on performance in each setting, so we refrain from computing a rank that takes into account both but enable the reader to do so via the tables containing all of the results in A.9.
> >
> > 11. About the eigenspectrum analysis. Why is it the case that the closer the distribution of eigenvalues is to uniform, the more balanced the attention profile of the recurrent weights is across dimensions of the eigen-transformed? And why does this have implications for the dimensionality of recurrent state-space dynamics?
> >
> > As we discussed in the comment addressed to all reviewers, we have now modified this to a singular value spectrum analysis (trends in the eigenspectrum show the same things, but the justification for using the singular values over the eigenvalues is stronger in our opinion). So we will address your questions in this context instead.
> >
> > * Consider an SVD on $W_{rec}$ given by $U \Sigma V^T$. If we examine $U \Sigma V^T h_t$, since U and V are unitary matrices, the expansion that occurs in the transformation induced by $W_{rec}$  can be isolated to singular value matrix $\Sigma$. So, it is not the case of distribution of singular values being closer to uniform the more balanced the attention profile of the recurrent weights is across dimensions of the hidden state (we have clarified this in the paper). Rather, the slower the singular values decay, the more evenly the expansion is balanced across basis directions which places the transformed vector into a space with higher effective dimensionality. In contrast, if the singular values decay faster, then fewer directions are needed to explain the variance of the transformed hidden state and thus it lies in a lower dimension. This has implications on robustness, as explained in the comment addressed to all reviewers.
> >
> > 12. In Figure 5, 6, 7 etc, what does the colored region mean? Is it the variance? Is it a confidence interval? If there are only three networks per parameter setting shouldn't they all be shown?
> >
> > The colored region is 1 SE (clarified in image captions). Yes, the three networks are all known, but we present SE as this is canonical for significance testing.
> >
> > 13. Is it really counterintuitive that with increasing sparsity, the dimensionality of the recurrent trajectories increases? Isn't sparsity just functioning as effectively increasing the rank?
> >
> > Yes, we believe that it is counterintuitive. Firstly, increasing rank increases the number of parameters while increasing sparsity decreases the number of parameters. Secondly, while increasing both sparsity and rank cause the singular values to decay slower, they do so in different fashions. In particular, we show that increasing sparsity reduces spectral radius/norm which reduces our attention profile across time whereas increasing rank increases the radius/norm which increases our attention profile across time. These distinctions are precisely what distinguishes pruning by increasing sparsity and pruning by reducing rank which is a key finding of our work.
> >
> > 14. It is not clear why higher dimensional dynamics would lead to more robust networks, as claimed in Figure 16. This seems to be in disagreement with the claims about constraining the network to be low-rank improved robustness in Section 4.3. Shouldn't a higher spectral norm contribute to higher dimensional dynamics? For a low spectral norm the network would quickly collapse onto a low-dimensional manifold. How do you explain then the RNNs have higher spectral norm and lower dimensionality in their dynamics?
> >
> > Apologies, this was a typo in the Figure 16 (now Figure 18). The dynamics in RNNs are indeed higher than the dynamics present in CfCs.
> >
> > 15. How is  delta_W calculated in Figure 7? What are the parameters used for the Adam optimizer?
> >
> > Both have been clarified in paper.

---

> > > ### Author Response · Authors · 2023-11-21
> > > **Response to reviewer 59d7 continued**
> > >
> > > 16. What number of parameters do the different networks have? In particular, how many more parameters do Cfcs have as a result of having two vanilla RNNs in them and another gating mechanism? Be more clear about how F  is parameterized. How could the increase in the number of parameters explain the increase in performance?
> > >
> > > We clarify in A.3 how F is parameterized and add a table in A.12 with the parameter counts of the models. The CfC has the same number of parameters as a GRU by virtue of the F network. We perform an analysis on the time constant network F in A.2 and leave further exploration regarding the reason for which it improves robustness to future work.
> > >
> > > 17. The last claim of Theorem 1 is only proved for orthogonal initialization?
> > >
> > > Yes, this was a typo in the statement of the theorem and has been resolved.
> > >
> > > Thanks again for your thorough review of our work! We hope that our revisions have resolved many of the concerns you brought to our attention.

---

> > > > ### Comment · Reviewer_59d7 · 2023-11-23
> > > >
> > > > Thank you for the response. I appreciate the additional details and the new version is taking steps in the right direction.
> > > >
> > > > My main concern is still with the interpretability of the (relative) performance of the different networks. Without seeing what the networks actually do, it is difficult to judge what the robustness analyses actually measure. Therefore, I will keep my score as it is.
> > > >
> > > > As a minor remark, even though sparsity decreases the number of parameters, I do believe it effectively increases the rank.

---

> ### Author Response · Authors · 2023-11-23
> **Response to reviewer 59d7**
>
> Thank you for your response. We address your points below.
>
> 1. My main concern is still with the interpretability of the (relative) performance of the different networks. Without seeing what the networks actually do, it is difficult to judge what the robustness analyses actually measure. Therefore, I will keep my score as it is.
>
> We clarify the following regarding your concern with the interpretability of the network performance and the robustness analyses.
>
> Regarding the interpretability of the performance, we maintain that the statistical significance of the online rewards (both in-distribution and under distribution) are evidence of the efficacy of changes in the model architecture and/or recurrent connectivity depending on the setting. The scope of this work is understanding how we can construct robust agents in the closed-loop setting. The canonical metric to do so is measuring the rewards collected by these agents, which we do in both Atari games and rigid body control tasks (Mujoco) which are two distinct environments. The results holding across both ALEs and the Mujoco environment across multiple ranks/sparsities (i.e. for low-rank trends we observe improved robustness in both rank 1 and rank 5 networks) is in our opinion sufficient evidence of the efficacy of the proposed parameterization. Furthermore, measuring robustness to various types of noise is a standard shift of data distribution in research and finds many applications in practice (i.e. Tobin et. al 2017, Domain Randomization for Transferring Deep Neural Networks from Simulation to the Real World).
>
> Regarding what the robustness analyses measure, we reiterate that the motivation of analyses is described thoroughly in Section 3. In particular, the recurrent gradients analysis (which is measuring the very similar things as the Lyapunov analysis you proposed) measures robustness across time (i.e how long does a perturbation affect the agent into the future). And the analysis on the spectral norm and the rate of decay of the singular values measures the robustness of the agent at a given point in time (i.e. how much does the perturbation I apply to the input affect agent's decision at time $t$). The spectral norm is a canonical metric for robustness, since the spectral norm of a weight matrix is itself its Lipschitz constant (which in turn effects the Lipschitz constant of the full network). Optimally, you would measure the Lipschitzness of the full network, but doing so is NP-hard. The decay of the singular values is motivated from the perspective of effective dimensionality, which is then supported via the state-space trajectory analysis. We want the trajectory to be expressible in fewer dimensions, since that means the distribution shift can only affect the trajectory in a few directions.
>
> 2. As a minor remark, even though sparsity decreases the number of parameters, I do believe it effectively increases the rank.
>
> Yes, we agree that while sparsity decrease the number of parameters, it increases the effective rank. This is precisely captured via the singular values decaying slower. However, note the distinction between increasing rank and decreasing sparsity in the context of our parameterization. **In particular, increasing sparsity in our parameterization reduces both the spectral radius and norm. In contrast, increasing rank in our parameterization increases both the spectral radius and the norm.**
>
> This means that while they both exhibit a slower decay in their singular value spectrum (and thus higher dimensionalities in state-space), they differ with respect to their attention across time. **Namely, models with high rank attend to more distant observations than models with high sparsity.** This is why we have two notions of robustness: robustness across time and robustness at a given point in time. Our analyses disentangle the two and explore how we can modulate our robustness in each of these cases as a function of rank and sparsity.

---

### Official Review · Reviewer_p6DM · 2023-11-01

**Soundness:** 3 good
**Presentation:** 1 poor
**Contribution:** 3 good
**Rating:** 6
**Confidence:** 2

**Summary:**

Paper analyzes how the sparsity and rank of recurrent connectivities effects the robustness of using these models for closed-loop control.

**Strengths:**

Originality
- The setting that this paper analyzes appears to be novel

Quality
- Paper presents a variety of in-depth analysis experiments, which seek to understand the effect of rank and sparsity on different aspects of the model

**Weaknesses:**

- In its current form, I found it a bit challenging to parse the main contributions of the paper. I think the reason might be that Section 3 (Parameterization of Connectivity) only details the form of the proposed connectivity and the theoretical ramifications on spectral radius and norm, without any mention of the main empirical findings in the paper. It is not until section 4 (Experiments), where the paper mentions the specific findings within each subsection. Even there, I found it difficult to get concrete takeaways from the experiments, because the results are usually describe in great detail without a high-level point. For example, in Sec 4.1, it seems like low-sparsity, high-rank CfCs and LSTMs are good in online settings, and LSTMs tend to be better than CfCs and RNNs at high sparsities, and low-rank, sparsity CfCs tend to be good under distribution shift. From reading this section, it was not clear to me which configuration of sparsity, rank, and architecture was most effective? In general, I think it would be helpful to distill the main takeaways from the experiments and incorporate them into Section 3, before explaining the details of the experimental setup and results.
- Right now, all of the neural network models used in the experiments use some kind of temporal connection. However, I believe this is not the standard architecture used when solving the tasks used in the experiments (Seaquest, halfcheetah, etc.). It would be helpful to to include an additional baseline using standard architectures (conv net, fully connect, etc) to get a sense for the return that a "vanilla" approach can get, and to better appreciate the significance of the robustness gains made by the additional sparsity and low rank formulations.

**Questions:**

See weaknesses section

---

> ### Author Response · Authors · 2023-11-21
> **Reviewer p6DM response**
>
> We thank the reviewer for taking the time to read and critique our work. We respond to your points below.
>
> 1. In its current form, I found it a bit challenging to parse the main contributions of the paper. Even there, I found it difficult to get concrete takeaways from the experiments, because the results are usually describe in great detail without a high-level point.
>
> We tend to agree with the reviewer (and note a similar sentiment shared by reviewer XFJy) that the previous version of the paper was challenging to parse. To remedy this, we have done the following:
> * Added references to the section in which each of the bullet points in the Introduction are shown (the bullet points represent the main contributions of the paper)
> - Simplified the structure of the paper as follows
>   - Section 3 motivates the parameterization of connectivity
>   - Section 4.1 describes the performance of the models at a high-level and makes note of the particularly interesting trends
>   - Section 4.2 focuses on the understanding the trends in performance of the in-distribution rewards
>   - Section 4.3 focuses on the understanding the trends in performance of the  rewards under distribution shift and juxtaposes it to the findings in-distribution
>   - Section 4.4 attempts to understand why the prior is most effective in CfCs
>   - Placed many of the secondary details into the appendix to avoid distracting the reader (i.e time constant analysis in A.2 and task dimension commentary in A.3)
> Added one/two sentences summaries of what the section aims to accomplish at the start of the section
> I.e. “Here, we aim to gain intuition for the results we observed in the online, in-distribution setting and in particular offer an explanation as to why the only effective form of pruning in-distribution was inducing sparsity in LSTMs.” (start of Section 4.2)
>
>
> 2. I think the reason might be that Section 3 (Parameterization of Connectivity) only details the form of the proposed connectivity and the theoretical ramifications on spectral radius and norm, without any mention of the main empirical findings in the paper. In general, I think it would be helpful to distill the main takeaways from the experiments and incorporate them into Section 3, before explaining the details of the experimental setup and results.
>
> We maintain that it is easier to parse the paper if the details regarding the parameterization and its implications on the spectral norm, spectral radius and singular value decay are separate from the analysis we do on the trained networks. This is primarily because the parameterization is motivated as a prior induced at initialization, whereas the subsequent empirical analysis are performed on the trained networks and extend beyond simply computing the spectral statistics we modulate via recurrent rank and sparsity. Furthermore, we want to separate the theoretical results at initialization from the empirical results on the trained networks.
>
> 3. It is not until section 4 (Experiments), where the paper mentions the specific findings within each subsection. For example, in Sec 4.1, it seems like low-sparsity, high-rank CfCs and LSTMs are good in online settings, and LSTMs tend to be better than CfCs and RNNs at high sparsities, and low-rank, sparsity CfCs tend to be good under distribution shift. From reading this section, it was not clear to me which configuration of sparsity, rank, and architecture was most effective?
>
> We have rewritten this section to make the main takeaways more clear. However, we do note that there is no one correct configuration since there is no one model that performs best in-distribution and under distribution shift. The general takeaway is that with respect to model types, LSTMs and CfCs tend to be most effective in-distribution and CfCs tend to be most effective under distribution shift (however, the fact that sparse LSTMs perform well in-distribution also counts for something because it is doing so with fewer parameters). And with respect to recurrent connectivity, sparsity presents itself as being effective in-distribution in LSTMs and low-rank presents itself as being effective across many models under distribution shift, most notably CfCs.
>
> 4. Right now, all of the neural network models used in the experiments use some kind of temporal connection. However, I believe this is not the standard architecture used when solving the tasks used in the experiments (Seaquest, halfcheetah, etc.). It would be helpful to include an additional baseline using standard architectures (conv net, fully connect, etc) to get a sense for the return that a "vanilla" approach can get.
>
> Our apologies for the confusion. We misstated that we used a TCN when in fact the architecture is actually a CNN applied to each point in time, hence there is no recurrent supervision. This has been corrected in the paper.
>
> Thanks again for your review of our work; we hope to have resolved your concerns!

---

> > ### Comment · Reviewer_p6DM · 2023-11-22
> > **Response**
> >
> > I appreciate the response from the authors. In the current form, I still find it a bit challenging to extract the main takeaways from the experiments section, so I will keep my score the same. However, I will note that recurrent connectivity is not my main area of research, and the feedback from other reviewers might be more informative.

---

> > > ### Author Response · Authors · 2023-11-22
> > > **Responding to p6DM**
> > >
> > > Thank you for the response. Would the reviewer be able to refer to specific paragraphs and/or structural details that they found confusing to parse? We are happy to change these if necessary.

---

### Author Response · Authors · 2023-11-21
**Summary of revisions**

We thank all the reviewers for their thoughtful criticisms and suggestions. Here, we address the main points raised by reviewers and outline the changes we have made to the paper in response. More specific points are addressed in the response to each of the individual reviewers.

Experimental additions

We have added/modified the following analyses in the paper:

1. Instead of analyzing the eigenspectrum as a proxy for the effective dimensionality of the transformation induced by the recurrent weights, we instead analyze the singular value spectrum. While they both show the same trends, we believe that analyzing the singular values has more merit. This is because if we have $W = U \Sigma V^T$, we can restrict scope of the transformation induced by $W$ to the singular values as the $U, V$ matrices are unitary. More details on this line of reasoning can be found in Section 3 and Appendix A.5.2.

2. We formalize the analysis on the singular value spectrum by measuring the rate of decay of the singular values, which we quantify by sorting the singular values and then normalized by the largest one. We also add this decay metric to Section 3 and discuss how it, along with spectral radius and spectral norm, changes as a function of the rank and sparsity of the recurrent connectivity.
In addition to analyzing the singular values of the recurrent weights, we also analyze the singular values of the input weights, as they too are relevant measures of model robustness.

3. Addressing points brought up by the reviewers, we added in an analysis on the recurrent Jacobians (Section 4.2) in order to verify the viability of spectral radius as a proxy for gradient evolution across time.

Structural modifications

To address concerns with readability, we have restructured the paper as follows:

1. In Section 3, we more thoroughly motivate the three spectral properties we care about. Then, we tie these properties to Theorem 1 and understand how these properties change as a function of rank and sparsity. We also clarify the claims we prove and the claims we only provide empirical evidence for.

2. Section 4.1 has mostly remained the same, wording was modified to make narrative more clear.

3. Section 4.2 has been rewritten in two aspects:

* We no longer say “vanishing gradient” (since this may have been implied an analysis during learning), we instead motivate the spectral radius as modulator of “recurrent-memory horizon”.

* The focus of section 4.2 remains on the analysis of network attention across time, however is motivated using the in-distribution results and then juxtaposed again the distribution shift results which flows into the next section.

4. Section 4.3 has been modified as follows:

* We have consolidated Sections 4.3 and 4.4 into a single section and instead motivate this section by discussing the distribution shift results
* The section can be decomposed into the following components, which are proxies for model robustness:
* Analysis of the spectral norm of the weights
* Analysis on the rate of decay of the singular values
* Analysis on the dimensionality of the model state-space trajectories

5. Section 4.4 (previously section 4.5) remains the same

6. Appendix A.5 contains a thorough description motivating the analyses we perform on the weight spectra, connecting those to various network dynamics

7. At a higher-level, we have rewritten our discussion of results to separately emphasize the two relevant axes of comparison
      1. Within a fixed model, how do things change as a function of rank and sparsity
      2. Within a fixed (rank, sparsity), how do things change as a function of the model type

---

> ### Author Response · Authors · 2023-11-22
> **Summary of second revision**
>
> We submit a revised version of the manuscript that reduces the file size of the Appendix, which has enabled us to merge the main paper and appendix into a single submission, which we hope improves readability. Since we cannot delete supplementary material, note that the Appendix is still contained there but is redundant as the contents there are the same as what is found in the main submission.

---

> > ### Author Response · Authors · 2023-11-23
> > **Final revision**
> >
> > We fixed some minor typos, but other than that everything else is unchanged.

---

### Meta-Review · Area_Chair_ZTKj · 2023-11-30

**Metareview:**

**Summary**:
The paper investigates the robustness of recurrent-based agents on non-stationary environments. The paper focuses on how sparsity and rank of different recurrent neural networks (RNNs) impact network dynamics, both theoretically and empirically. The main funding is that smaller spectral radius and norm can yield better generalization under distribution shift. The paper also shows that closed-form continuous-time neural networks are more amenable to a low-rank and sparse connectivity prior than vanilla RNNs and LSTMs.

**Strengths**: Reviewers appreciated that the proposed theory is "simple and insightful" and lauded the thoroughness of the experiments. They noted that the experiments seem reproducible, and liked that the paper combined ideas from different areas (dynamical systems, neuroscience, deep learning).

**Weaknesses**: The main weakness reviewers noted was in the organization of the paper. The authors made significant revisions to the structure to the paper during the rebuttal in attempts to address this. There were also some suggestions about presenting the results (e.g., reward normalization), which were also incorporated into the revised paper.

**Justification For Why Not Higher Score:**

Even after a very long discussion and multiple revisions to the paper, the reviewers still had some open questions about the paper. One reviewer still found the paper a bit difficult to read.

**Justification For Why Not Lower Score:**

The paper seems technically correct, combines ideas from a few different technical areas, and is accompanied by thorough experimental results.

---

### Decision · Program_Chairs · 2024-01-16

Accept (spotlight)